# Statistical Test for Saliency Maps of Graph Neural Networks via Selective Inference

**Shuichi Nishino**                                    *nishino.shuichi.nagoyaml@gmail.com*
*Nagoya University*
*RIKEN*

**Tomohiro Shiraishi**                                    *shiraishi.tomohiro.nagoyaml@gmail.com*
*Nagoya University*
*RIKEN*

**Teruyuki Katsuoka**                                    *katsuoka.teruyuki.nagoyaml@gmail.com*
*Nagoya University*

**Ichiro Takeuchi**                                    *takeuchi.ichiro.n6@f.mail.nagoya-u.ac.jp*
*Nagoya University*
*RIKEN*

**Reviewed on OpenReview:** *https://openreview.net/forum?id=5NkXTCVa7F*

## Abstract

Graph Neural Networks (GNNs) have gained prominence for their ability to process graph-structured data across various domains. However, interpreting GNN decisions remains a significant challenge, leading to the adoption of saliency maps for identifying salient subgraphs composed of influential nodes and edges. Despite their utility, the reliability of GNN saliency maps has been questioned, particularly in terms of their robustness to input noise. In this study, we propose a statistical testing framework to rigorously evaluate the significance of saliency maps. Our main contribution lies in addressing the inflation of the Type I error rate caused by double-dipping of data, leveraging the framework of *Selective Inference*. Our method provides statistically valid *p*-values while controlling the Type I error rate, ensuring that identified salient subgraphs contain meaningful information rather than random artifacts. The method is applicable to a variety of saliency methods with piecewise linearity (e.g., Class Activation Mapping). We validate our method on synthetic and real-world datasets, demonstrating its capability in assessing the reliability of GNN interpretations.

## 1 Introduction

Graph Neural Networks (GNNs) have gained considerable attention as a powerful approach for analyzing data with an inherent graph structure. Graph structures, defined by nodes and edges, appear in various types of data, including social networks, molecular structures, 3D scans, and spatiotemporal data. The flexibility of graph representation enables the utilization of rich structural information, which is difficult to process using conventional vector-based representations. As a result, GNNs have achieved notable success in applications ranging from social network analysis to climate forecasting and bioinformatics.

A significant challenge in applying GNNs is to interpret their decision-making processes. To ensure interpretability, it is essential to identify and visualize which parts of the input graph—such as specific nodes or subgraphs—contribute most significantly to the model's prediction. A well-known approach to this problem extends saliency-based techniques originally developed for computer vision, such as Class Activation Mapping (CAM) (Simonyan et al., 2013) and Grad-CAM (Selvaraju et al., 2017), to the graph domain (Pope et al., 2019). These methods produce saliency maps that highlight influential components of a graph; from

these, one can extract salient subgraphs that may reflect characteristic structure in the input data. However, several studies have raised concerns about their reliability, making it crucial to quantify the reliability (Li et al., 2024b;a). This is especially important in mission-critical applications (e.g., medical diagnosis) and scientific discovery (e.g., neuroscience).

In this study, we propose a novel statistical test for evaluating the reliability of saliency maps of GNNs. We focus on discovering characteristic structures hidden in the input data using saliency maps. Instead of blindly trusting the values of saliency maps, we treat them as hypotheses and assess their statistical significance. Our method quantifies the reliability in the form of $p$-values. $P$-values indicate the statistical significance of a saliency map, allowing us to assess whether the observed patterns are meaningful or occur merely by chance. In other words, it enables us to control the Type I error rate (i.e., false positive rate) at a predefined significance level.

In this paper, as a proof of concept, we consider a simple GNN for electroencephalography (EEG) data. EEG signals are recorded as spatiotemporal voltage changes across the scalp and have recently been increasingly analyzed using GNNs (Demir et al., 2021; Lin et al., 2023; Klepl et al., 2024). In particular, we identify the spatiotemporal locations where stimulus-induced voltage shifts occur, known as Event-Related Potentials (ERPs)[1] in neuroscience. Specifically, when a GNN is trained to classify stimulus types from EEG data, it can reveal characteristic spatiotemporal patterns in brain activity by extracting the salient subgraph that underlies its classification decision. Figure 1 illustrates the workflow of our method as applied to EEG data. First, we transform the multidimensional time-series data into a graph representation. Next, we apply a trained GNN and use a saliency map to extract the salient subgraph. Finally, we evaluate the statistical significance of the extracted salient subgraph and quantify its reliability through $p$-values. Our proposed statistical testing framework enables a rigorous assessment of whether the observed voltage shift constitutes a statistically significant change, thereby identifying the regions and time periods where meaningful differences occur.

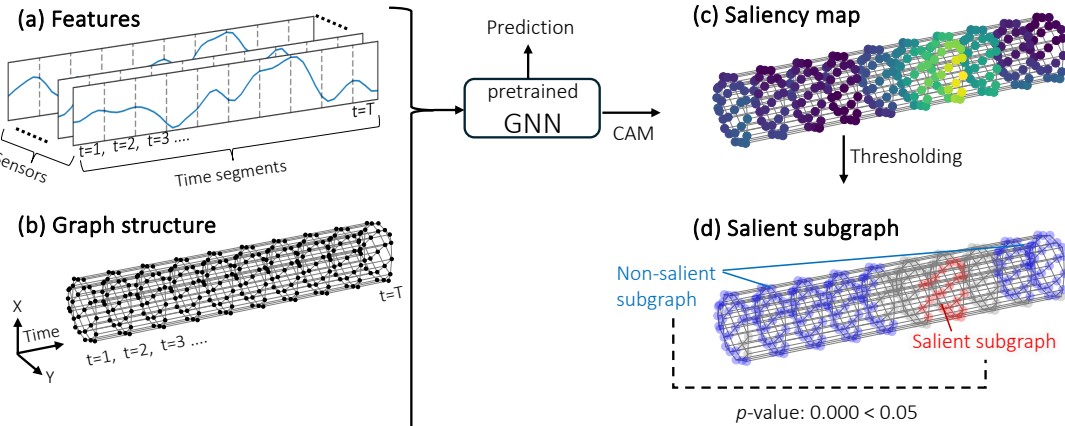

Figure 1: The analysis workflow for computing a saliency map from graph data and assessing its statistical significance. (a) The time-series data from each sensor are segmented along the temporal axis, forming nodes with features extracted from the resulting segments (sensor count × number of time segments). (b) A graph structure is constructed by defining edges based on the spatial adjacency of sensors and the temporal continuity of the segments. (c) A trained GNN, combined with a saliency estimation method (e.g., CAM), is applied to compute the saliency of each node. (d) By applying two different thresholds—upper and lower—to the resulting saliency map, we obtain a salient subgraph (red) and a non-salient subgraph (blue), while the remaining nodes are shown in gray. In this example, certain spatial regions become salient at two different time steps. Finally, we perform statistical hypothesis testing and compute a $p$-value. In this example, $p < 0.05$ indicates statistical significance.

---

[1]ERP is a well-known class of stimulus-induced voltage shifts, widely used in neuroscience and cognitive science to study brain responses, cognitive processing, and sensory perception. They offer key insights into neural dynamics and have applications in clinical diagnostics, brain-computer interfaces, and cognitive neuroscience.

To the best of our knowledge, this study is the first to propose a rigorous statistical testing framework for assessing the reliability of GNN saliency maps. A fundamental challenge in conducting such statistical tests is that the same dataset is used both to select the salient subgraph and to perform the test itself. This practice violates the standard statistical assumption that the hypothesis should be specified independently of the data used for testing. This issue is widely known as *double-dipping* in the statistical literature (Breiman, 1992; Kriegeskorte et al., 2009; Benjamini, 2020). When EEG signals contain noise, the saliency map tends to highlight and extract false salient subgraphs that are affected by the noise and are not truly important. If we then test these subgraphs using the same EEG data (i.e., the data that contains the same noise), we risk accepting patterns that are in fact artifacts of the noise. In other words, the Type I error rate is inflated, a phenomenon often referred to as *selection bias*, and the results of statistical tests become unreliable. Note that this double-dipping problem is unavoidable, as saliency maps are designed to explain the model's prediction for a specific input instance, which prevents the use of separate datasets for estimation and testing. Therefore, we are inevitably required to test a *data-driven hypothesis*—the salient subgraph—using the same data that was used to select it.

We address this challenge by employing Selective Inference (SI), a statistical testing framework that has attracted significant attention over the past decade for testing data-driven hypotheses (Fithian et al., 2014; Loftus & Taylor, 2014; Lee et al., 2016). The fundamental principle of SI is to perform statistical testing while accounting for the data-driven selection of hypotheses. By incorporating the selection process into the analysis, SI effectively mitigates the inflation of Type I error caused by data-driven selection. In this work, we analyze the selection mechanism underlying GNN saliency maps and incorporate it into statistical testing via the SI framework. The proposed statistical test yields valid *p*-values and enables reliable evaluation of the saliency maps and properly controll the Type I error rate at the nominal significance level.

As a proof of concept for statistical testing on GNN saliency maps, we focus on a standard GNN architecture with CAM, a fundamental method for saliency computation. However, the proposed framework is applicable to a wide range of GNN architectures and saliency methods that satisfy piecewise linearity. Piecewise linearity is a property that arises in various graph architectures, such as Graph Convolutional Networks (GCNs) and Graph Isomorphism Networks (GINs) (Xu et al., 2019) with ReLU activation, as well as in gradient-based saliency methods, including Grad-CAM, applied to these architectures. We also assume that the input graphs are attributed graphs, where each node is associated with continuous feature vectors, as in the EEG example described above. A more detailed discussion of limitations and scopes is provided in Section 5.

**Related work.** Interpretable machine learning has received increasing attention in GNNs (Yuan et al., 2022). Among various approaches, saliency-based approaches have emerged as a prominent category. Saliency-based methods that infer importance for input features are known as feature-based methods, with Class Activation Mapping (CAM) and Grad-CAM being the most well-known examples. These methods were first introduced in the context of Convolutional Neural Networks to identify important input regions (Simonyan et al., 2013; Selvaraju et al., 2017). CAM for GNNs (Pope et al., 2019) extends CAM to graph data by using node feature activations to identify influential nodes and edges. Its extension, Grad-CAM for GNNs (Pope et al., 2019), incorporates gradient information to make it applicable to a broader range of GNN models. Other feature-based saliency map methods include GraphLIME (Huang et al., 2022), which uses local surrogate models for feature-level explanations, and GraphSVX (Duval & Malliaros, 2021), which applies Shapley value approximations to quantify node and edge contributions. In these methods, sets of highly influential nodes can often be interpreted as salient subgraphs. Alternatively, subgraph-based methods, including GNNExplainer (Ying et al., 2019) and SubgraphX (Yuan et al., 2021), directly infer such structures. However, several studies (Li et al., 2024b;a) have highlighted the fragility of interpretability methods for GNNs, since small perturbations to the input graph can significantly alter saliency, thus reducing their reliability. To our knowledge, this is the first study to propose a principled statistical framework for assessing the reliability of GNN saliency maps. Statistical significance alone is not sufficient for interpretability.[2] However, it enables us to use saliency maps to discover underlying patterns in the input data while controlling the Type I error rate.

---

[2] Our method can be viewed as a complementary metric of interpretability; see Section 5.

Over the past decade, SI has been actively studied to offer statistical tests for data-driven hypothesis selection. SI was initially developed to evaluate the reliability of feature selection in linear models (Fithian et al., 2014; Tibshirani et al., 2016; Loftus & Taylor, 2014; Suzumura et al., 2017; Le Duy & Takeuchi, 2021; Sugiyama et al., 2021; Duy & Takeuchi, 2022) and later extended to other problem settings (Lee et al., 2015; Choi et al., 2017; Neufeld et al., 2022; Shiraishi et al., 2024a; Matsukawa et al., 2024). In the context of deep learning, SI was first introduced by Duy et al. (2022) and later expanded for CNNs (Miwa et al., 2023; 2024; Katsuoka et al., 2024; 2025), transformers (Shiraishi et al., 2024a) and RNNs (Shiraishi et al., 2024b). Despite growing interest in explainability for GNNs, rigorous statistical testing for GNN interpretations remains an open challenge. In this study, we propose a novel statistical test for GNN saliency maps, which performs statistical test conditioned on the selected salient subgraph using the SI framework.

**Contributions.** Our contributions can be summarized as follows:

- We propose a framework to quantify the statistical significance of GNN saliency within the context of statistical testing. This framework enables the quantitative evaluation of the reliability of GNN saliency in the form of $p$-values. In our framework, we extract a salient subgraph and non-salient subgraph from the saliency map and perform a statistical test to evaluate its difference (see Section 3).

- We propose an SI-based approach to obtain statistically valid $p$-values for the aforementioned statistical test. Our method is an exact (non-asymptotic) inference method that works well even with a small sample size. We leverage the piecewise linear structure of GNN to compute the $p$-values efficiently (see Section 4).

- We conducted experiments on synthetic and real-world datasets, through which we show that our proposed method can control the Type I error rate and provides good results in practical applications. Our code is available at `https://github.com/ni-shu/si_for_gnn` (see Section 6).

## 2 Problem Setup

In this section, we briefly explain the GNN, its saliency map method and the extraction of salient subgraphs.

### 2.1 GNNs and Saliency Maps

First, we define the graph data as follows:
$$G_X = (X, A),$$
where $A \in \{0, 1\}^{n \times n}$ is the adjacency matrix with $n$ nodes, and $A_{ij} = 1$ if there is an edge between nodes $i$ and $j$, and $A_{ij} = 0$ otherwise. $X = (\boldsymbol{x}_1, \boldsymbol{x}_2, \ldots, \boldsymbol{x}_n)^\top \in \mathbb{R}^{n \times d}$ is the accumulated feature vector of all nodes in the graph, where $\boldsymbol{x}_i \in \mathbb{R}^d$ is the feature vector of node $i$. In the case of EEG data in Section 1, $\boldsymbol{x}_i$ represents a segment of the time series, and the adjacency matrix $A$ is constructed based on the spatial adjacency of sensors and the temporal continuity of the segments.

As a proof of concept for statistical testing on GNN saliency maps, we focus on a standard GNN architecture with CAM, a fundamental method for saliency computation. Figure 2 illustrates a standard architecture of a GNN equipped with CAM (Pope et al., 2019). CAM is a widely used technique for interpreting neural network decisions by identifying input regions most influential to the output. In the context of GNNs, we adopt a dual-branch architecture that enables both graph-level prediction and node-wise saliency computation. Specifically, a stack of GCN layers first encodes the input graph into node embeddings. These embeddings are then processed through two distinct paths: one aggregates node features using global average pooling followed by a linear classifier; the other applies CAM. The CAM layer computes node-wise scores by applying the transposed classifier weights to the node embeddings. The approach of CAM enables efficient saliency estimation without additional training overhead. We show the details of the GCN layer and CAM layer in Appendix A.

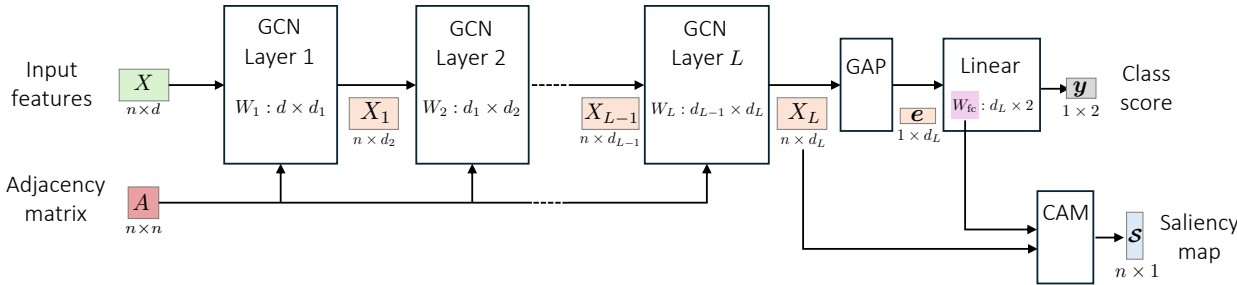

Figure 2: Architecture of a GNN equipped with CAM. The input consists of a node feature matrix $X$ and an adjacency matrix $A$. Multiple GCN layers perform neighborhood aggregation and feature transformation. The resulting node embeddings are processed in two parallel branches: one for graph-level classification via global average pooling and a linear layer, and the other for saliency computation using CAM. The CAM branch computes class-wise node relevance scores by applying the transposed classifier weights to the node embeddings.

## 2.2 Salient Subgraphs

Our goal is to evaluate the reliability of saliency maps generated by GNNs. As the output of the CAM layer, we obtain a saliency map $\mathcal{S}(G_X) \in [0,1]^n$ where each node $i$ is assigned a contribution score $\mathcal{S}_i(G_X) \in [0,1]$, representing its importance in the model's prediction. To evaluate the reliability of $\mathcal{S}(G_X)$, we define the salient subgraph and non-salient subgraph and investigate whether they have statistically significant differences. From $\mathcal{S}(G_X)$, we identify the set of nodes $\mathcal{V}_X^+$ contributing to the GNN classification. This set can be determined as follows with an arbitrary threshold $\tau_u \in [0,1]$:

$$\mathcal{V}_X^+ = \{i \in [n] \mid \mathcal{S}_i(G_X) > \tau_u\} \in 2^{[n]}. \tag{1}$$

We refer to the subgraph induced by $\mathcal{V}_X^+$ as the salient subgraph, as it contains the most influential nodes in the model's prediction. Similarly, the non-salient subgraph is defined using the threshold $\tau_l \in [0,1]$ for $\tau_l < \tau_u$:

$$\mathcal{V}_X^- = \{i \in [n] \mid \mathcal{S}_i(G_X) \leq \tau_l\} \in 2^{[n]}. \tag{2}$$

For simplicity, we denote the set of salient and non-salient subgraphs as

$$\mathcal{V}_X = \left\{\mathcal{V}_X^+, \mathcal{V}_X^-\right\}. \tag{3}$$

Our goal is to quantify the statistical significance of graph saliency by evaluating the feature-wise differences between the salient node sets $\mathcal{V}_X^+$ and the non-salient node sets $\mathcal{V}_X^-$ as $p$-values. A statistically significant difference indicates that the observed saliency region has some meaningful interpretation. In contrast, a lack of statistical significance suggests that the observed saliency region may simply be due to noise and could be meaningless.

# 3 A Statistical Testing Framework for GNN Saliency

In this section, we present our first contribution: a framework for quantifying the statistical significance of GNN saliency within the context of statistical testing. This framework allows for the quantitative assessment of GNN saliency reliability through p-values.

## 3.1 Assumptions

We formulate a statistical test to quantify the reliability of saliency maps generated by GNNs. For simplicity, we introduce the accumulated feature vector of all nodes in the graph as $\boldsymbol{X} = (\boldsymbol{x}_1^\top, \boldsymbol{x}_2^\top \ldots, \boldsymbol{x}_n^\top)^\top \in \mathbb{R}^{nd}$. To formulate the problem as a statistical test, we assume that feature vector $\boldsymbol{X}$ is a random vector drawn from the following statistical model:

$$\boldsymbol{X} = \boldsymbol{\mu} + \boldsymbol{\varepsilon}, \ \boldsymbol{\varepsilon} \sim \mathcal{N}(\boldsymbol{0}, \Sigma), \tag{4}$$

where $\boldsymbol{\mu} = (\boldsymbol{\mu}_1^\top, \boldsymbol{\mu}_2^\top, \ldots, \boldsymbol{\mu}_n^\top)^\top \in \mathbb{R}^{nd}$ is the unknown true signal and $\boldsymbol{\varepsilon} \in \mathbb{R}^{nd}$ is the Gaussian noise with covariance matrix $\Sigma$.[3] The model in (4) above is general in that it makes no prior structural assumptions about the signal in the feature vectors, while assuming normality for the noise (see Appendix E.1 for experiments verifying the robustness against violations of the normality assumption).

Given a trained GNN model, we can obtain a saliency map $\mathcal{S}(G_{\boldsymbol{X}}) \in [0,1]^n$ and salient subgraph $\mathcal{V}_{\boldsymbol{X}}^+$ and non-salient subgraph $\mathcal{V}_{\boldsymbol{X}}^-$ by (1) and (2).

### 3.2 Statistical Test

To assess the significance of the salient subgraph, we formulate the problem as a hypothesis test to compare the mean feature values between the salient subgraph $\mathcal{V}_{\boldsymbol{X}}^+$ and the non-salient subgraph $\mathcal{V}_{\boldsymbol{X}}^-$. Specifically, we consider the following null hypothesis is $H_0$ and the alternative hypothesis is $H_1$:

$$H_0 : \frac{1}{|\mathcal{V}_{\boldsymbol{X}}^+|} \sum_{v \in \mathcal{V}_{\boldsymbol{X}}^+, i \in [d]} \mu_{v,i} = \frac{1}{|\mathcal{V}_{\boldsymbol{X}}^-|} \sum_{u \in \mathcal{V}_{\boldsymbol{X}}^-, j \in [d]} \mu_{u,j}$$

$$\text{vs.} \tag{5}$$

$$H_1 : \frac{1}{|\mathcal{V}_{\boldsymbol{X}}^+|} \sum_{v \in \mathcal{V}_{\boldsymbol{X}}^+, i \in [d]} \mu_{v,i} \neq \frac{1}{|\mathcal{V}_{\boldsymbol{X}}^-|} \sum_{u \in \mathcal{V}_{\boldsymbol{X}}^-, j \in [d]} \mu_{u,j},$$

where $\mu_{v,i}$ is the $i$-th element of vector $\boldsymbol{\mu}_v$, which denotes the true signal of the $i$-th feature of node $v$. The null hypothesis $H_0$ assumes that there is no difference in the mean feature values between the two subgraphs, implying that the selection of salient nodes does not correspond to any meaningful distinction in feature space. The alternative hypothesis $H_1$ indicates that the saliency map captures meaningful characteristics of the graph. Note that both $H_0$ and $H_1$ are dependent on data $\boldsymbol{X}$, i.e. they are data-driven hypotheses. Unlike traditional hypothesis testing, where hypotheses are fixed in advance, these hypotheses behave probabilistically. In Section 3.4, we introduce the SI framework as a valid testing approach for such data-driven hypotheses.

A natural way to define the test statistic, which quantifies the difference stated in (5), is as follows:

$$T(\boldsymbol{X}) = \frac{1}{|\mathcal{V}_{\boldsymbol{X}}^+|} \sum_{v \in \mathcal{V}_{\boldsymbol{X}}^+, i \in [d]} x_{v,i} - \frac{1}{|\mathcal{V}_{\boldsymbol{X}}^-|} \sum_{u \in \mathcal{V}_{\boldsymbol{X}}^-, j \in [d]} x_{u,j}. \tag{6}$$

The test statistic of (6) measures the discrepancy between the mean feature values of the salient subgraph and the non-salient subgraph. Note that the test statistic can be rewritten as $T(\boldsymbol{X}) = \boldsymbol{\eta}^\top \boldsymbol{X}$, where $\boldsymbol{\eta} \in \mathbb{R}^{nd}$ is a weight vector (see Appendix B for the detailed definition). For convenience, we use a normalized version of this statistic without loss of generality:

$$T(\boldsymbol{X}) = \frac{\boldsymbol{\eta}^\top \boldsymbol{X}}{\sqrt{\boldsymbol{\eta}^\top \Sigma \boldsymbol{\eta}}}. \tag{7}$$

To perform the hypothesis test, we first determine the sampling distribution of the test statistic $T(\boldsymbol{X})$ under the null hypothesis $H_0$, and then compute the $p$-value as the probability of obtaining the observed statistic (or a more extreme one) under this distribution. If the $p$-value is less than the significance level $\alpha \in (0,1)$ (e.g., 0.05), we reject $H_0$ and conclude that the salient subgraph is significantly different from the non-salient subgraph. Our goal is to compute the $p$-value which satisfies

$$\mathbb{P}_{H_0} (p \leq \alpha) = \alpha, \ \forall \alpha \in (0,1). \tag{8}$$

The property in 8 means that the Type I error rate is controlled at any significance level $\alpha \in (0,1)$, and the test is said to be *valid*. A statistical test is said to be *valid* if the $p$-values obtained by the test satistifies the property in (8).

---

[3]We assume that the covariance matrix $\Sigma$ is known for simplicity and discuss the scenario when we use sample estimate of the covariance matrix in Appendix E.2.

### 3.3 Challenges in Computing Valid $P$-Values

A key challenge in performing this test lies in the fact that the hypothesis is *data-driven*—that is, the null hypothesis $H_0$ depends on the data $\boldsymbol{X}$. To compute the $p$-value, which quantifies the extremeness of the observed value under $H_0$, we need to fix $H_0$ in some principled way.

Here, we describe a method referred to as the *naive method*. Although this method is not valid as a statistical test, it serves as a useful contrast to motivate our proposed approach. In the naive method, the randomness of $H_0$ is ignored. It does not account for the fact that the salient subgraph is selected based on the data. Therefore, it assumes that the null distribution of the test statistic $T(\boldsymbol{X})$ follows the standard normal distribution. Using this null distribution, the $p$-value for a given observed value $\boldsymbol{X}^{\mathrm{obs}}$ is defined as

$$p_{\mathrm{naive}} = \mathbb{P}_{\mathrm{H}_0} \left( |T(\boldsymbol{X})| > |T(\boldsymbol{X}^{\mathrm{obs}})| \right). \tag{9}$$

However, in reality, treating $H_0$ as non-random is not valid, so the naive method does not guarantee the required property in (8), leading to an inflated Type I error rate. Intuitively, this means that the saliency map may be overly emphasized, and the naive method may incorrectly judge it as statistically significant. As a result, the test statistic, which is defined based on difference between the two subgraphs, is more likely to be large, leading to an increased rejection rate of the null hypothesis. This issue is known as *selection bias* in the SI literature (Lee et al., 2016). As shown in Section 6, our experimental results demonstrate that selection bias leads to an inflated Type I error rate, exceeding the specified significance level. Our main contribution is to propose a statistical test that resolve this issue based on the SI framework.

### 3.4 Concept of Selective Inference

We introduce the framework of SI, also known as post-selection inference, to compute valid $p$-values. The key idea behind SI is to perform conditional hypothesis testing, where the inference is conditioned on the *selection event* that the hypothesis $H_0$ is selected based on the data. In our setting, this corresponds to conditioning on the selected subgraphs:

$$T(\boldsymbol{X}) \mid \{\mathcal{V}_{\boldsymbol{X}} = \mathcal{V}_{\boldsymbol{X}^{\mathrm{obs}}}\}. \tag{10}$$

The conditioning in (10) means that we only consider the $\boldsymbol{X}$ whose subgraph $\mathcal{V}_{\boldsymbol{X}}$ is the same as the observed one $\mathcal{V}_{\boldsymbol{X}^{\mathrm{obs}}}$. Conditioning on the selection event allows us to treat $H_0$ as fixed. The *selective $p$-value* computed under this conditional framework represents the probability that the observed test statistic $T(\boldsymbol{X}^{\mathrm{obs}})$ could have arisen by chance, given that the selection event has occurred and the null hypothesis holds. By quantifying graph saliency based on the selective $p$-value, we can control the Type I error at a desired significance level.

## 4 Selective $P$-Values for Graph Saliency

In this section, we present our second contribution: a method for computing valid selective $p$-values that appropriately quantify the statistical significance of graph saliency in the context of the statistical testing problem introduced in the previous section.

### 4.1 Definition of Selective $P$-Values

The parameters of conditional sampling distribution (10) contain not only the parameter of interest, which is related to the test statistic, but also the nuisance parameter. We can eliminate the nuisance parameter by conditioning on its sufficient statistic, which is defined as

$$\mathcal{Q}_{\boldsymbol{X}} = \left( I - \frac{\Sigma \boldsymbol{\eta} \boldsymbol{\eta}^\top}{\boldsymbol{\eta}^\top \Sigma \boldsymbol{\eta}} \right) \boldsymbol{X}. \tag{11}$$

The elimination of nuisance parameters by conditioning is a standard approach in statistics for deriving the conditional sampling distribution (Lehmann et al., 1986). Existing studies on SI have also conditioned by the same sufficient statistic $\mathcal{Q}_{\boldsymbol{X}}$ in (11) (Fithian et al., 2014; Lee et al., 2016). So, the conditional test statistic for computing the selective $p$-value is defined as

$$T(\boldsymbol{X}) \mid \{\mathcal{V}_{\boldsymbol{X}} = \mathcal{V}_{\boldsymbol{X}^{\mathrm{obs}}}, \mathcal{Q}_{\boldsymbol{X}} = \mathcal{Q}_{\boldsymbol{X}^{\mathrm{obs}}}\}. \tag{12}$$

Finally, the selective $p$-value is defined as

$$p_{\mathrm{selective}} = \mathbb{P}_{\mathrm{H}_0} \left( |T(\boldsymbol{X})| > |T(\boldsymbol{X}^{\mathrm{obs}})| \mid \boldsymbol{X} \in \mathcal{X} \right), \tag{13}$$

where the conditional data space $\mathcal{X}$ is defined as

$$\mathcal{X} = \left\{ \boldsymbol{X} \in \mathbb{R}^{nd} \mid \mathcal{V}_{\boldsymbol{X}} = \mathcal{V}_{\boldsymbol{X}^{\mathrm{obs}}}, \mathcal{Q}_{\boldsymbol{X}} = \mathcal{Q}_{\boldsymbol{X}^{\mathrm{obs}}} \right\}. \tag{14}$$

### 4.2 Property of Selective $P$-Values

The selective $p$-value in (13) satisfies the following theorem.

**Theorem 1.** *Under the null hypothesis* $\mathrm{H}_0$ *in* (5)*, for any* $\alpha \in (0,1)$*, the selective $p$-value in* (13) *satisfies the propery in* (8)*, i.e.,*

$$\mathbb{P}_{\mathrm{H}_0} \left( p_{\mathrm{selective}} \leq \alpha \right) = \alpha.$$

This is a well-known property of the $p$-value derived through SI. The proof of Theorem 1 in our setting is given in Appendix C.2. This theorem means that test procedure based on the selective $p$-value in (13) controls the Type I error rate at any significance level $\alpha \in (0,1)$. Note that this theorem do not require asymptotic discussions to guarantee the validity of the selective $p$-value. In Section 6, we experimentally demonstrate that the selective $p$-value can control the Type I error rate even when the sample size (i.e., the number of nodes $n$ and the number of features $d$) is finite. Furthermore, we emphasize that we make no assumptions on the training data or training process of the GNN. Our method guarantees control of the Type I error rate even when the GNN is trained on ill-conditioned data, such as datasets containing a substantial number of false positive examples.

### 4.3 Computation of Selective $P$-Values

To compute the selective $p$-value in (13), we must characterize the conditional data space $\mathcal{X}$ in (14), which corresponds to all feature vectors $\boldsymbol{X}$ that yield the same selected subgraph $\mathcal{V}_{\boldsymbol{X}}$ as the observed one $\mathcal{V}_{\boldsymbol{X}^{\mathrm{obs}}}$. This requirement can be viewed as an inverse problem for GNN, and is particularly challenging due to the complexity of the GNNs forward computation and decision process. We address this challenge based on the following two key insights. First, the computation of CAM scores can be formulated as a piecewise linear function for a broad class of GNN architectures (see Lemma 1). Second, leveraging this piecewise linearity, the selective $p$-value can be computed exactly by reducing the problem to a one-dimensional search (see Lemma 2), which is solvable using techniques called *parametric programming* (Duy & Takeuchi, 2022).

**Piecewise linearity of GNN saliency maps.** The first key observation is that the saliency map computed by CAM is a piecewise linear function of the input features. This structure enables us to analytically characterize the conditional data space $\mathcal{X}$ in terms of linear constraints.

**Lemma 1.** *For the network architecture defined in Figure 2 with ReLU activation function, the saliency map* $\mathcal{S}(G_{\boldsymbol{X}})$ *is a piecewise linear function of* $\boldsymbol{X}$*. That is, the input space* $\mathbb{R}^{nd}$ *can be partitioned into a finite set of convex polytopes* $\{\mathcal{R}_k\}_{k=1}^K$ *for some* $K \in \mathbb{N}$ *such that in each region* $\mathcal{R}_k$*,* $\mathcal{S}(G_{\boldsymbol{X}})$ *is an affine function of* $\boldsymbol{X}$*:*

$$\forall \boldsymbol{X} \in \mathcal{R}_k, \ \forall i \in [n], \quad \mathcal{S}_i(G_{\boldsymbol{X}}) = C_i^{(k)} \boldsymbol{X} + b_i^{(k)},$$

*where* $C_i^{(k)} \in \mathbb{R}^{n \times nd}$ *and* $b_i^{(k)} \in \mathbb{R}^n$ *are region-specific coefficients.*

The proof is omitted since it follows from standard results.[4] In fact, many standard neural network components—such as linear layers, convolutional layers, and max-pooling operations—are also piecewise linear. Therefore, the piecewise linearity of the CAM output holds not only for GCNs but also for a wide range of GNN architectures. This implies that our proposed framework can be readily extended to various types of GNNs beyond those explicitly considered in this paper as a proof of concept.

**One-dimensional search.** The conditional data space $\mathcal{X}$ consists of a union of convex polytopes, due to the piecewise linearity of the saliency map with respect to the input $\boldsymbol{X} \in \mathbb{R}^{nd}$. However, directly characterizing or sampling from $\mathcal{X}$ is computationally difficult because of the high dimensionality of $\boldsymbol{X}$. To overcome this, we leverage the structure of the conditional data space to reduce the problem to a one-dimensional search along a carefully chosen linear path in the input space.

**Lemma 2.** *The set $\mathcal{X}$ as defined in* (14) *can be rewritten using a scalar parameter $z = T(\boldsymbol{X}) \in \mathbb{R}$ as*

$$\mathcal{X} = \left\{ \boldsymbol{X}(z) \in \mathbb{R}^{nd} \mid \boldsymbol{X}(z) = \boldsymbol{a} + \boldsymbol{b}z, \quad z \in \mathcal{Z} \right\},$$

*where*

$$\boldsymbol{a} = \mathcal{Q}_{\boldsymbol{X}^{\mathrm{obs}}} \in \mathbb{R}^{nd}, \quad \boldsymbol{b} = \Sigma\boldsymbol{\eta}/\sqrt{\boldsymbol{\eta}^\top \Sigma \boldsymbol{\eta}} \in \mathbb{R}^{nd},$$

*and the truncation intervals $\mathcal{Z}$ is given by*

$$\mathcal{Z} = \left\{ z \in \mathbb{R} \mid \mathcal{V}_{\boldsymbol{a}+\boldsymbol{b}z} = \mathcal{V}_{\boldsymbol{X}^{\mathrm{obs}}} \right\}.$$

The proof of Lemma 2 is provided in Appendix C.1. This lemma shows that the conditional data space $\mathcal{X}$ lies entirely on a one-dimensional subspace of $\mathbb{R}^{nd}$, parameterized by the scalar parameter $z$. In our setting, $\mathcal{X}$ is the subset of graph data that generates the same salient subgraphs as those of the observed data. Consequently, it suffices to restrict our attention to a one-dimensional subset, which enables efficient and exact computation of the selective $p$-value. The remaining challenge is to identify $\mathcal{Z}$.

**Parametric programming.** To identify the truncation intervals $\mathcal{Z}$, we employ a divide-and-conquer strategy. Specifically, we assume that we have a procedure to compute the interval $[L_z, U_z]$ for any $z \in \mathbb{R}$ which satisfies: for any $r \in [L_z, U_z]$, the subgraphs $\mathcal{V}_{\boldsymbol{a}+\boldsymbol{b}r}$ and $\mathcal{V}_{\boldsymbol{a}+\boldsymbol{b}z}$ are the same, i.e.,

$$\forall r \in [L_z, U_z], \ \mathcal{V}_{\boldsymbol{a}+\boldsymbol{b}r} = \mathcal{V}_{\boldsymbol{a}+\boldsymbol{b}z}.$$

Then the truncation intervals $\mathcal{Z}$ can be obtained as the union of the intervals $[L_z, U_z]$ as

$$\mathcal{Z} = \bigcup_{z \in \mathbb{R} \ \mathrm{s.t.} \mathcal{V}_{\mathrm{a+bz}}=\mathcal{V}_{\mathrm{X^{obs}}}} [L_z, U_z]. \tag{15}$$

The aggregation procedure in (15) is known as *parametric programming*, originally introduced to SI field by Duy & Takeuchi (2022). Building on the concept of parametric programming, the problem of identification of $\mathcal{Z}$ is divided into subproblems on how to compute the interval $[L_z, U_z]$.

Each subproblem is tractable due to the piecewise linearity of the saliency map. From Lemma 1 and 2, it follows that $\mathcal{S}(G_{\boldsymbol{X}(z)})$ is a piecewise linear function of $z$. Consequently, the thresholding operations used to define the salient and non-salient subgraphs in (1) and (2) can be expressed as a system of linear inequalities with respect to $z$. The explicit formulation of these inequalities is provided in Appendix D.1. In more practical cases, we may want to perform filtering or normalization on the saliency map $\mathcal{S}(G_{\boldsymbol{X}(z)})$ before thresholding. Many filters (e.g., Gaussian filters) preserve linearity, so the same formulation can be applied. When applying a threshold to a saliency map normalized to the range $[0, 1]$, a slightly different formulation is required (see Appendix D.2). We emphasize that our framework conducts selective inference with consideration of the thresholding process and ensures control of the Type I error rate for any specified threshold.

By solving these linear inequality systems, we can identify each interval $[L_z, U_z]$, and by aggregating them using (15), we obtain the full truncation set $\mathcal{Z}$. The complete procedure for computing the selective $p$-value via parametric programming is summarized as Algorithm 1, provided in Appendix D.3.

---

[4]The composition of piecewise linear functions is also piecewise linear. Therefore, it suffices to verify that each layer in the GNN (e.g., linear, convolutional, pooling, ReLU) is piecewise linear to conclude that the overall CAM computation is piecewise linear.

# 5   Limitation and Scope

In this section, we discuss the scope of the framework and its current limitations from several aspects.

**GNN architectures and saliency methods.**   In this study, as a proof of concept, we employed a standard GCN architecture along with the conventional saliency method, CAM. However, our framework is not limited to this specific network architecture or saliency method. As discussed in Section 4, the proposed method is applicable when the operations within a NN can be expressed as piecewise linear functions. Although this requirement may appear restrictive at first glance, most operations commonly employed in GCNs are either inherently linear or can be represented by piecewise linear functions. A notable exception arises when nonlinear activation functions, such as the sigmoid function, are used; however, since sigmoid functions can be approximated with arbitrary precision by piecewise linear segments, selective $p$-values can still be computed with high accuracy. This property similarly holds for GINs. When applying CAM, Grad-CAM, Grad & GradInput (Shrikumar et al., 2017) to these networks, the resulting saliency maps also satisfy the piecewise linearity property, making the proposed method applicable. In the experiments in Section 6, we applied the proposed method to the GCN with CAM and confirmed that it controlled the Type I error rate. In addition, we applied Grad-CAM, Grad, and GradInput to both GCNs and GINs and obtained similar results, which are reported in Appendix G. In contrast, architectures such as Graph Transformer Networks (Yun et al., 2019) and Graph Attention Networks (Veličković et al., 2018), or saliency methods such as GNNExplainer or GraphLIME, contain components that cannot be readily expressed or approximated as piecewise linear functions. Some gradient-based methods, such as Integrated Gradients (IG) (Sundararajan et al., 2017), are also not piecewise linear, posing challenges for the direct application of the proposed framework.

**Noise distribution.**   Our data generation model in (4) is general in the sense that no assumptions are made regarding the true signal features $\boldsymbol{\mu}$. On the other hand, our SI framework builds on the normality of the noise, as is the case in other existing SI studies. We experimentally investigated the robustness when the noise deviates from the normal distribution (see Appendix E.1), and found that the type I error can be controlled almost at the nominal significance level. From a theoretical perspective, one possible direction to address this issue is to introduce an asymptotic theory (Tian & Taylor, 2018; 2017; Markovic et al., 2017). However, the computation of selective $p$-values based on asymptotic theory becomes much more complex, and the computational methods introduced in Section 4 of this paper cannot be directly applied.

**Computational complexity.**   The computational cost of selective $p$-value calculation using the algorithm proposed in Section 4 depends on the complexity of the GNN architecture and the size of the data. As the GNN architecture becomes more complex, the number of segments in the piecewise linear function increases. Similarly, as the data size (i.e., the number of samples and features) grows, the event associated with the selection of the saliency region becomes more intricate, resulting in a larger number of segments. An increase in the number of segments leads to more iterations in the *while* loop of Algorithm 1, thereby proportionally increasing the computational cost. Although the computational cost did not pose a bottleneck in the GNNs and datasets used in our numerical experiments, computational strategies such as parallelization may need to be considered for larger networks and datasets.

**Choice of test statistic.**   In this paper, we quantified the statistical significance of the GNN saliency by considering the null and alternative hypotheses defined in (5) and employing the test statistic presented in (6). However, our proposed framework is not limited to this specific test statistic. As detailed in Section 3.2, the framework is applicable as long as the test statistic can be expressed as a linear function $\boldsymbol{\eta}^\top X$, where $\boldsymbol{\eta} \in \mathbb{R}^{nd}$ is a vector that depends on the data $X$. Although we believe the test statistic adopted in this study is reasonable, depending on the perspective from which graph saliency is evaluated, it may be worthwhile to consider alternative test statistics that better capture different aspects of saliency.

**Thresholding strategy.**   In this paper, we proposed a framework that controls the Type I error rate for any fixed threshold. Our method is also applicable when the range of saliency values is normalized before applying the threshold, for cases where the range of the saliency distribution varies depending on the data. In contrast, when adaptively adjusting the threshold according to the characteristics of the saliency distribution,

it is necessary to perform SI that accounts for this process. One promising direction is to introduce a new conditioning scheme, inspired by existing studies on selective inference for image segmentation (Tanizaki et al., 2020), which discuss conditioning on thresholding processes that consider the distribution of pixel intensities in image data.

**Interpretability.** Here, we further extend the discussion to consider the application of selective $p$-values to a comprehensive evaluation of interpretability. From one perspective, the proposed method can be regarded as a statistical means to check *plausibility* of an obtained saliency map—that is, the degree to which it conforms to expert knowledge. Since the data-generating model in (4) and the null and alternative hypotheses in (5) can be viewed as a mathematical model of such domain knowledge, the proposed method quantifies the plausibility of the explanation in the form of $p$-values with respect to this model. However, the interpretability of machine learning models is a multifaceted concept. Several evaluation criteria, such as *fidelity*, *consistency*, and *completeness*, explicitly focus on whether the explanation accurately reproduces the model's behavior. Moreover, criteria such as *simplicity* assess human readability. Since the proposed method does not directly assess these aspects, challenges remain in employing selective $p$-values as part of a comprehensive evaluation of interpretability. One possible direction to address this challenge is to use the proposed test as a complementary evaluation metric, in conjunction with existing evaluation metrics for interpretability, such as the entropy-based sparsity score of GNN saliency distributions (Funke et al., 2022). Even when a generated explanation lacks interpretability, our testby Theorem 1correctly controls the Type I error rate. Thus, the $p$-value serves as a complementary indicator to existing interpretability metrics, helping to guarantee the reliability of the saliency map. The optimal way to integrate the proposed test with existing metrics remains a topic for future work. For example, a gating strategy could be usedbefore proceeding to interpretability evaluation, apply our test to admit only statistically significant subgraphs as candidate explanations, thereby filtering out false positives due to incidental noise in advance. Weighted aggregation of metrics is another possibility. Complementary use with existing metrics is expected to enhance the robustness and credibility of interpretability assessments.

# 6 Numerical Experiments

In this section, we compare the proposed method with other methods and demonstrate that the proposed method exhibits high power (true positive rate) while controlling the Type I error rate (false positive rate) below the significance level compared to other methods. First, experiments are conducted on synthetic datasets, followed by similar experiments on EEG datasets. The architecture of the GNN and the saliency map used in the experiments are shown in Figure 2. We set the number of GCN layers $L = 3$ and the number of hidden units $d_l = 10$ for $l \in [L]$. We normalized the saliency map to the range $[0, 1]$ before applying thresholds, which were set to $\tau_l = 0.3$ and $\tau_u = 0.7$. All experiments were conducted with a significance level of $\alpha = 0.05$.

## 6.1 Methods for Comparison

In our experiments, we compare the proposed method (`proposed`) with three other methods: `naive`, `w/o-pp`, and `Bonferroni`.

- `naive`: This method uses a classical $z$-test without conditioning: that is, we compute the naive $p$-value as described in Equation (9).

- `Bonferroni`: This is a method to control the Type I error rate by using the Bonferroni correction. There are $3^n$ ways to choose the subgraphs $\mathcal{V}_{\boldsymbol{X}}$. We then compute the Bonferroni-corrected $p$-value as $p_{\text{bonferroni}} = \min(1, 3^n \cdot p_{\text{naive}})$.

- `w/o-pp`: An ablation study that excludes the parametric programming technique described in (15).

## 6.2 Synthetic Data Experiments

**Setup.** To evaluate the Type I error rate, we varied the number of features ($d \in \{5, 10, 15, 20\}$) and the number of nodes ($n \in \{32, 64, 128, 256\}$). If not specified, we used $d = 5$ and $n = 256$. For each setting, we statistically tested 1,000 graphs $G_{\boldsymbol{X}} = (\boldsymbol{X}, A)$, which are generated using the following procedure. The adjacency matrix $A \in \mathbb{R}^{n \times n}$ is generated by randomly adding edges such that the average degree of each node is 3. We also generated a null feature vector $\boldsymbol{X} \sim \mathcal{N}(\boldsymbol{0}, \Sigma)$, where the covariance matrix $\Sigma \in \mathbb{R}^{nd \times nd}$ is defined as the Kronecker product $\Sigma = \Sigma_{\text{space}} \otimes \Sigma_{\text{feature}}$. We considered two types of covariance matrices:

- **Independence**: $\Sigma_{\text{space}} = I_n$ and $\Sigma_{\text{feature}} = I_d$.
- **Correlation**: $\Sigma_{\text{space}}$ is defined by $(\Sigma_{\text{space}})_{ij} = 0.1^{d_{ij}}$, where $d_{ij}$ is the shortest path length between nodes $i$ and $j$ in a graph, and $\Sigma_{\text{feature}}$ is defined by $(\Sigma_{\text{feature}})_{kl} = 0.1^{|k-l|}$.

To evaluate the power, we varied the signal strength ($\delta \in \{1.0, 1.5, 2.0, 2.5\}$). For each setting, we iterated 1,000 experiments. In each iteration, we generated a feature vector $\boldsymbol{X} \sim \mathcal{N}(\boldsymbol{\mu}, \Sigma)$, where $\boldsymbol{\mu}$ is obtained by flipping each element of the $\boldsymbol{0} \in \mathbb{R}^{nd}$ to $\delta$ with probability 0.1. The adjacency matrix $A$ is generated in the same way as in the Type I error rate experiments. The covariance matrix $\Sigma$ follows the same two settings as in the Type I error rate experiments.

**Results.** The results are shown in Figure 3. The `proposed`, `w/o-pp`, and `Bonferroni` methods successfully control the Type I error rate at the nominal significance level, even under finite sample sizes, whereas the `naive` method fails due to selection bias. Since `naive` failed to control the Type I error rate, we excluded it from the power analysis. Among the remaining methods, `proposed` achieves the highest power across varying signal strengths. See Appendix E.1 for results under non-Gaussian settings, and Appendix E.2 for results using estimated covariance matrices. We also confirmed that proposed method can control the Type I error rate for any combinations of thresholds $\tau_l$ and $\tau_u$ (see Appendix F).

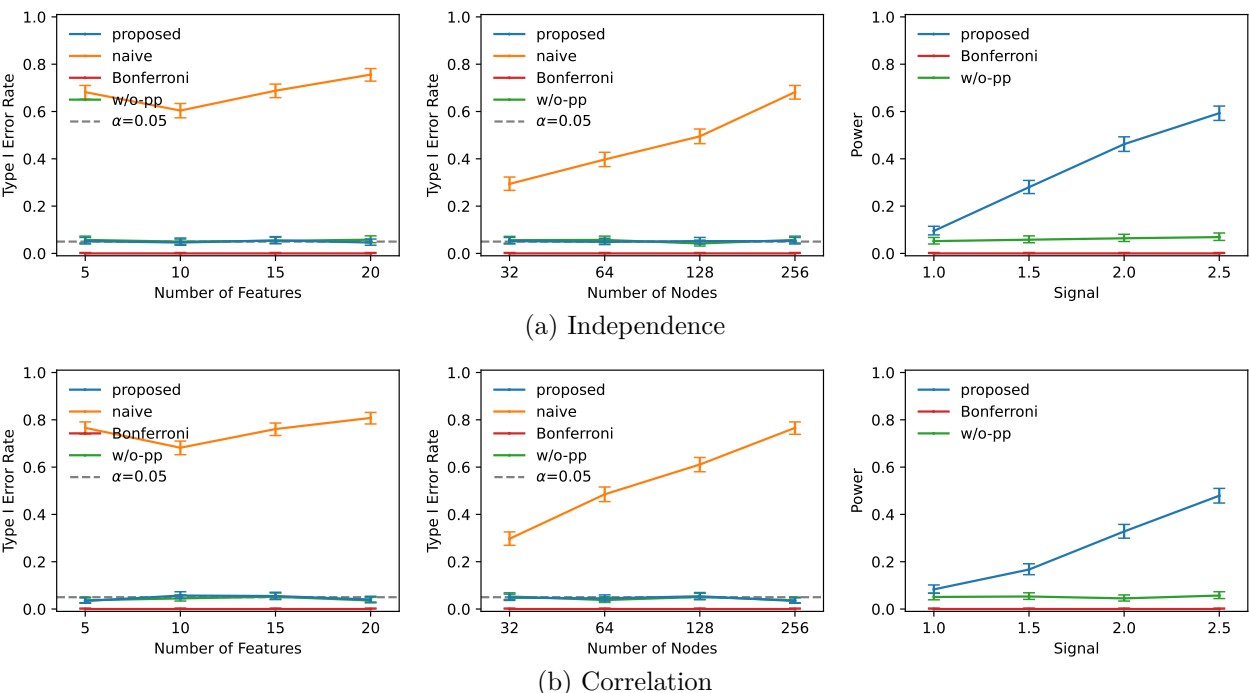

Figure 3: Results on synthetic data. The left and middle columns show the Type I error rates with varying numbers of features and nodes, respectively. The `proposed` and `w/o-pp` methods successfully control the Type I error rate at the $\alpha = 0.05$ level, with nearly overlapping curves, while the `naive` method fails to do so. The results of `Bonferroni` are almost zero, because it is too conservative. The figures in the right column show the power with varying signal strength. The `proposed` method has the highest power.

### 6.3 Real Data Experiments

**Setup.** We evaluated the effectiveness of our proposed method in detecting spatiotemporal characteristics in multi-dimensional time-series data using a dataset which contains known characteristics. We used the EEG Dataset provided by Won et al. (2022), which contains EEG signals recorded during the well-known Rapid Serial Visual Presentation (RSVP) task in neuroscience. This dataset contains two categories of EEG signals: positive and negative. Each sample corresponds to one-second segments of multidimensional EEG signals recorded after the presentation of visual stimuli. The positive data are known to contain a stimulus-induced positive potential shift, whereas the negative data are known to show little to no such shift. To demonstrate the effectiveness of the proposed framework, we applied it to the task of identifying the spatiotemporal characteristics of the evoked EEG response, specifically discovering the sensor locations and latencies at which the potential shift occurs. The positive potential observed in the positive data is referred to as the P300, which typically emerges approximately 300 to 600 milliseconds after stimulus presentation. It is well-established that the P300 manifests predominantly over parietal and frontal scalp regions. In this experiment, we investigated whether our proposed method yields results consistent with these established findings.

The dataset contains 55 participants, each with 40 positive and 560 negative samples, resulting in a total of 33,000 EEG samples. Each sample consists of recordings from 28 EEG sensors capturing the scalp-wide potential distribution, with each sensor measuring 50 time points corresponding to the first second after stimulus onset. To represent the data as a graph, each time series was segmented into non-overlapping windows of length 5, resulting in a total of $28 \times 10$ segments. Each segment was treated as a node in the graph representation of the EEG data, where each node encapsulates temporal features from a specific EEG sensor. Edges between nodes were defined based on spatial or temporal adjacency: an edge was added if two nodes originated from the same sensor in consecutive time windows, or from adjacent sensors at the same time step. For further details on the dataset and preprocessing steps, see Appendix H.1. For training the GNN model, we used EEG data from 15 participants. For the remaining 40 participants, we utilized 520 negative samples per participant for covariance estimation, and used the remaining samples as test data.

**Results.** The results of `proposed` and `naive` are shown in Figure 4 and 5. In the positive example, the saliency method successfully identifies the salient subgraph, which corresponds to the P300 component of the EEG signal. The $p$-values obtained from `naive` tend to be small even for the negative class EEG signals, indicating that they are not suitable for quantifying the reliability of salient subgraphs. In contrast, the $p$-values of `proposed` are generally large for positive data and small for negative data. This result suggests that `proposed` can effectively detect true positive cases while avoiding false positive detections. For more examples, see Appendix H.2. We also conducted experiments with modified real datasets, demonstrating that `proposed` can control the Type I error rate when the real dataset is modified to follow the null hypothesis assumptions; see Appendix H.3.

## 7 Conclusion

In this study, we proposed a novel statistical testing framework for assessing the reliability of GNN saliency maps by leveraging SI. Our framework computes statistically valid $p$-values, ensuring that extracted salient subgraphs reflect meaningful contributions rather than arising by chance. Through extensive experiments on synthetic and real-world datasets, we demonstrated that our method effectively mitigates Type I errors. As explainability in GNNs continues to be a crucial research area, our approach provides a rigorous statistical foundation for evaluating model interpretations, paving the way for more reliable and trustworthy GNN-based decision-making systems.

**Broader Impact**

The selective inference framework proposed in this study mathematically underpins the reliability of GNN saliency explanations by rigorously controlling the Type I error rate at any chosen threshold. This enables safer and more transparent operation of decision-making systems based on GNNs—for example, in health-care (such as EEG analysis and other biomedical data), weather and environmental modeling, or industrial

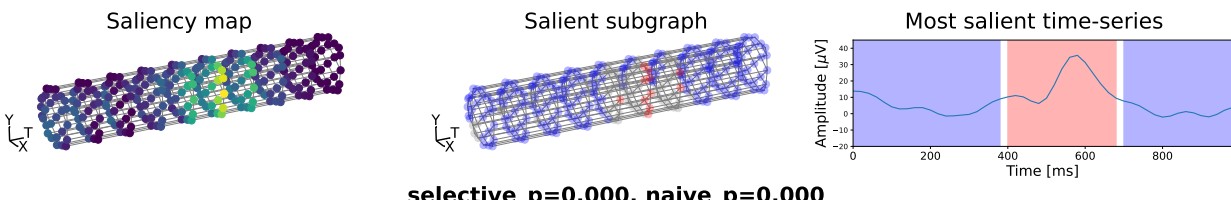

**selective_p=0.000, naive_p=0.000**

Figure 4: Positive example. The left figure shows the saliency map, where brighter nodes indicate higher importance. In this figure, the x-axis corresponds to the anterior direction of the scalp, while the y-axis corresponds to the rightward direction, providing a spatial interpretation of the saliency distribution. The middle figure shows the obtained subgraphs, where the red nodes are selected as salient subgraph and the blue nodes are selected as non-salient subgraph. The right figure shows the EEG signals of the most salient channel and its salient and non-salient segments (red and blue, respectively). CAM successfully identifies the segments of EEG signals that exhibit a potential shift. Notably, the identified salient segments correspond to positive deflections occurring approximately 300 600 milliseconds post-stimulus, predominantly over parietal and frontal scalp regions characteristics consistent with the well-documented features of the P300 component. Furthermore, the waveform morphology of the salient segment shown in the right figure closely resembles the pattern reported in Figure 6 of Won et al. (2022), further supporting the neurophysiological plausibility of the identified explanation. Below the example, we report the corresponding $p$-values, demonstrating that the proposed method correctly detects the positive sample.

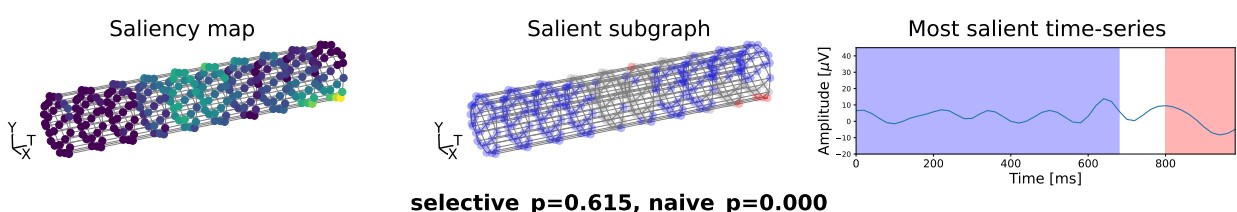

**selective_p=0.615, naive_p=0.000**

Figure 5: Negative example. See Figure 4 for the interpretation. Our selective $p$-value is sufficiently large, indicating correct exclusion of spurious saliency, whereas the naive $p$-value is misleadingly small.

anomaly detection—by mitigating decisions based on spurious or overly optimistic interpretations. In particular, noisy biomedical signals are known to carry the risk of false positive saliency leading to diagnostic errors, and our method helps extract only statistically significant patterns, providing clinicians, researchers, and engineers with trustworthy grounds for interpretation and supporting reliable explanations.

**Acknowledgments**

This work was partially supported by JST CREST (JPMJCR21D3, JPMJCR22N2), JST Moonshot R&D (JPMJMS2033-05), and RIKEN Center for Advanced Intelligence Project and RIKEN Junior Research Associate Program.

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

## A    Graph Neural Network Architecture

**GCN layer.**    The GCN layer is a layer that computes the output of the GNN by aggregating the information of the neighbors of each node. The GCN layer is defined as

$$X_l = \sigma\left(\tilde{A} X_{l-1} W\right),$$

where $l \in [L]$ is the layer index, $X_{l-1} \in \mathbb{R}^{n \times d_{l-1}}$ is the input of the layer, $X_l \in \mathbb{R}^{n \times d_l}$ is the output of the layer, and $W_{l-1} \in \mathbb{R}^{d_{l-1} \times d_l}$ is the weight matrix. $\sigma$ is the activation function, which is typically a nonlinear function such as ReLU. $\tilde{A} \in \mathbb{R}^{n \times n}$ is the normalized adjacency matrix which is defined as $\tilde{A} = D^{-1/2} A D^{-1/2}$, where $D = \mathrm{diag}(A\mathbf{1}) \in \mathbb{N}^{n \times n}$ is the degree matrix.

**CAM layer.**    The CAM layer is a layer that computes the saliency map, which is the contribution of each node to the decision of the GCN-based GNN. The use of the CAM layer assumes that the class score of the GNN uses the output of the GAP layer. The GAP layer is defined as

$$\boldsymbol{e} = \frac{1}{n} X_l{}^\top \mathbf{1},$$

where $X_l \in \mathbb{R}^{n \times d_l}$ is the output of the GCN layer just before the GAP layer, and $\boldsymbol{e} \in \mathbb{R}^{d_l}$ is the output vector of the GAP layer, which is the aggregation of the features of the entire graph for each feature dimension. The classification score of class $c$ is defined as

$$y_c = \boldsymbol{e}^\top \boldsymbol{w}_c,$$

where $\boldsymbol{w}_c \in \mathbb{R}^{d^{(l)}}$ is the weight vector of class $c$. The CAM layer utilizes $\boldsymbol{w}_c$ and the output of the GCN layer just before the GAP layer $X_l$ to compute the class $c$ saliency map, wchich is defined as

$$\sigma\left(X_l \boldsymbol{w}_c\right) \in \mathbb{R}^n,$$

where the activation function $\sigma$ is the ReLU function.

## B    Detailed Definition of the Test Statistic

The weighted vector $\boldsymbol{\eta} \in \mathbb{R}^{nd}$ in the test statistic $T(\boldsymbol{X}) = \boldsymbol{\eta}^\top \boldsymbol{X}$ in (7) is defined as follows:

$$\boldsymbol{\eta} = \left(\boldsymbol{c}_1(\boldsymbol{X})^\top, \boldsymbol{c}_2(\boldsymbol{X})^\top, \ldots, \boldsymbol{c}_n(\boldsymbol{X})^\top\right)^\top \in \mathbb{R}^{nd},$$

with component-wise definition:

$$\boldsymbol{c}_v(\boldsymbol{X}) = \begin{cases} \frac{1}{|\mathcal{V}_{\boldsymbol{X}}^+|} \mathbf{1} \in \mathbb{R}^d & \text{if } v \in \mathcal{V}_{\boldsymbol{X}}^+ \\ -\frac{1}{|\mathcal{V}_{\boldsymbol{X}}^-|} \mathbf{1} \in \mathbb{R}^d & \text{if } v \in \mathcal{V}_{\boldsymbol{X}}^- \\ \mathbf{0} \in \mathbb{R}^d & \text{otherwise.} \end{cases}$$

# C  Proof of Theorem

## C.1  Proof of Lemma 2

According to the conditioning on $\mathcal{Q}_{\boldsymbol{X}} = \mathcal{Q}_{\boldsymbol{X}^{\text{obs}}}$, we have

$$\mathcal{Q}_{\boldsymbol{X}} = \mathcal{Q}_{\boldsymbol{X}^{\text{obs}}} \iff \left(I_n - \frac{\Sigma \boldsymbol{\eta}\boldsymbol{\eta}^\top}{\boldsymbol{\eta}^\top \Sigma \boldsymbol{\eta}}\right)\boldsymbol{X} = \mathcal{Q}_{\boldsymbol{X}^{\text{obs}}} \iff \boldsymbol{X} = \boldsymbol{a} + \boldsymbol{b}z,$$

Then, we have

$$\begin{aligned}
\{\boldsymbol{X} \in \mathbb{R}^n \mid \mathcal{V}_{\boldsymbol{X}} = \mathcal{V}_{\boldsymbol{X}^{\text{obs}}}, \mathcal{Q}_X = \mathcal{Q}_{\boldsymbol{X}^{\text{obs}}}\} &= \{\boldsymbol{X} \in \mathbb{R}^n \mid \mathcal{V}_{\boldsymbol{X}} = \mathcal{V}_{\boldsymbol{X}^{\text{obs}}}, \boldsymbol{X} = \boldsymbol{a} + \boldsymbol{b}z, z \in \mathbb{R}\} \\
&= \{\boldsymbol{a} + \boldsymbol{b}z \in \mathbb{R}^n \mid \mathcal{V}_{\boldsymbol{a}+\boldsymbol{b}z} = \mathcal{V}_{\boldsymbol{X}^{\text{obs}}}, z \in \mathbb{R}\} \\
&= \{\boldsymbol{a} + \boldsymbol{b}z \in \mathbb{R}^n \mid z \in \mathcal{Z}\}.
\end{aligned}$$

## C.2  Proof of Theorem 1

Based on intervals set $\mathcal{Z}$ in Lemma 2, we can derive the null distribution of the test statistic in (12) as a truncated normal distribution. Specifically, under the null hypothesis $H_0$ in (5), the conditional test statistic in (12) follows the truncated standard normal distribution $TN(0, 1, \mathcal{Z})$ with mean 0 and standard deviation 1, where $\mathcal{Z}$ is the truncation intervals defined in Lemma 2. Thus, by probability integral transformation, we have

$$p_{\text{selective}} \mid \{\mathcal{V}_{\boldsymbol{X}} = \mathcal{V}_{\boldsymbol{X}^{\text{obs}}}, \mathcal{Q}_{\boldsymbol{X}} = \mathcal{Q}_{\boldsymbol{X}^{\text{obs}}}\} \sim \text{Unif}(0, 1),$$

which leads to

$$\mathbb{P}_{H_0}\left(p_{\text{selective}} \le \alpha \mid \mathcal{V}_{\boldsymbol{X}} = \mathcal{V}_{\boldsymbol{X}^{\text{obs}}}, \mathcal{Q}_{\boldsymbol{X}} = \mathcal{Q}_{\boldsymbol{X}^{\text{obs}}}\right) = \alpha, \quad \forall \alpha \in (0, 1).$$

For any $\alpha \in (0, 1)$, by marginalizing nuisance parameters, we have

$$\begin{aligned}
\mathbb{P}_{H_0}\left(p_{\text{selective}} \le \alpha \mid \mathcal{V}_{\boldsymbol{X}} = \mathcal{V}_{\boldsymbol{X}^{\text{obs}}}\right) &= \int \mathbb{P}_{H_0}\left(p_{\text{selective}} \le \alpha \mid \mathcal{V}_{\boldsymbol{X}} = \mathcal{V}_{\boldsymbol{X}^{\text{obs}}}, \mathcal{Q}_{\boldsymbol{X}} = \mathcal{Q}_{\boldsymbol{X}^{\text{obs}}}\right) \\
&\quad \times \mathbb{P}_{H_0}\left(\mathcal{Q}_{\boldsymbol{X}} = \mathcal{Q}_{\boldsymbol{X}^{\text{obs}}} \mid \mathcal{V}_{\boldsymbol{X}} = \mathcal{V}_{\boldsymbol{X}^{\text{obs}}}\right) d\mathcal{Q}_{\boldsymbol{X}^{\text{obs}}} \\
&= \alpha \int \mathbb{P}_{H_0}\left(\mathcal{Q}_{\boldsymbol{X}} = \mathcal{Q}_{\boldsymbol{X}^{\text{obs}}} \mid \mathcal{V}_{\boldsymbol{X}} = \mathcal{V}_{\boldsymbol{X}^{\text{obs}}}\right) d\mathcal{Q}_{\boldsymbol{X}^{\text{obs}}} \\
&= \alpha.
\end{aligned}$$

Therefore, we also obtain

$$\begin{aligned}
\mathbb{P}_{H_0}\left(p_{\text{selective}} \le \alpha\right) &= \sum_{\mathcal{V}_{\boldsymbol{X}^{\text{obs}}}} \mathbb{P}_{H_0}(\mathcal{V}_{\boldsymbol{X}^{\text{obs}}})\mathbb{P}_{H_0}\left(p_{\text{selective}} \le \alpha \mid \mathcal{V}_{\boldsymbol{X}} = \mathcal{V}_{\boldsymbol{X}^{\text{obs}}}\right) \\
&= \alpha \sum_{\mathcal{V}_{\boldsymbol{X}^{\text{obs}}}} \mathbb{P}_{H_0}(\mathcal{V}_{\boldsymbol{X}^{\text{obs}}}) \\
&= \alpha.
\end{aligned}$$

# D  Details of Algorithm

## D.1  How to Identify the Interval of $z$

From Lemma 1 about the piecewise linearity of the GNN saliency map and Lemma 2, we can see that the saliency map $\mathcal{S}(G_{\boldsymbol{X}(z)})$ is a piecewise linear function of $z$, i.e., $\forall r \in [L'_z, U'_z]$, $\forall i \in [n]$, $\mathcal{S}_i(G_{\boldsymbol{X}(r)}) = c_i + \beta_i r$. If we have the interval $[L'_z, U'_z]$, we can compute the interval $[L_z, U_z]$ by the following formula:

$$[L_z, U_z] = [L_z^+, U_z^+] \cap [L_z^-, U_z^-], \tag{16}$$

where $[L_z^+, U_z^+]$ and $[L_z^-, U_z^-]$ are the intervals $\mathcal{V}_{\boldsymbol{a}+\boldsymbol{b}z}^+$ is same and $\mathcal{V}_{\boldsymbol{a}+\boldsymbol{b}z}^-$ is same, respectively. Specifically, they are defined as

$$[L_z^+, U_z^+] = \bigcap_{i \in [n]} [L_z^+(i), U_z^+(i)],$$

$$[L_z^+(i), U_z^+(i)] = \begin{cases} \left[\max\left(L'_z, \frac{\tau_u - c_i}{\beta_i}\right), U'_z\right] & \text{if } \left(\beta_i > 0 \wedge i \in \mathcal{V}_{\boldsymbol{a}+\boldsymbol{b}z}^+\right) \vee \left(\beta_i < 0 \wedge i \notin \mathcal{V}_{\boldsymbol{a}+\boldsymbol{b}z}^+\right) \\ \left[L'_z, \min\left(U'_z, \frac{\tau_u - c_i}{\beta_i}\right)\right] & \text{if } \left(\beta_i < 0 \wedge i \in \mathcal{V}_{\boldsymbol{a}+\boldsymbol{b}z}^+\right) \vee \left(\beta_i > 0 \wedge i \notin \mathcal{V}_{\boldsymbol{a}+\boldsymbol{b}z}^+\right), \end{cases} \tag{17}$$

$$[L_z^-, U_z^-] = \bigcap_{i \in [n]} [L_z^-(i), U_z^-(i)],$$

$$[L_z^-(i), U_z^-(i)] = \begin{cases} \left[\max\left(L'_z, \frac{\tau_l - c_i}{\beta_i}\right), U'_z\right] & \text{if } \left(\beta_i > 0 \wedge i \notin \mathcal{V}_{\boldsymbol{a}+\boldsymbol{b}z}^-\right) \vee \left(\beta_i < 0 \wedge i \in \mathcal{V}_{\boldsymbol{a}+\boldsymbol{b}z}^-\right) \\ \left[L'_z, \min\left(U'_z, \frac{\tau_l - c_i}{\beta_i}\right)\right] & \text{if } \left(\beta_i < 0 \wedge i \notin \mathcal{V}_{\boldsymbol{a}+\boldsymbol{b}z}^-\right) \vee \left(\beta_i > 0 \wedge i \in \mathcal{V}_{\boldsymbol{a}+\boldsymbol{b}z}^-\right). \end{cases} \tag{18}$$

## D.2  Normalization of Saliency Map

In Appendix D.1, we consider the case where the saliency map is not normalized. Here, we consider the case where we want to normalize the saliency map before applying a threshold. With a slight abuse of notation, we denote the saliency of $i$-th node $\mathcal{S}_i(G_X)$ as $\mathcal{S}_i$. The normalized saliency $\mathcal{S}_i^{\text{norm}}$ is then defined as:

$$\mathcal{S}_i^{\text{norm}} = \frac{\mathcal{S}_i - \min(\mathcal{S}_i)}{\max(\mathcal{S}_i) - \min(\mathcal{S}_i)},$$

provided that $\max S_i \neq \min S_i$. Then subgraphs $\mathcal{V}_{\boldsymbol{X}}^+$ and $\mathcal{V}_{\boldsymbol{X}}^-$ are defined as

$$\mathcal{V}_{\boldsymbol{X}}^+ = \{i \in [n] \mid \mathcal{S}_i^{\text{norm}} > \tau_u\},$$
$$\mathcal{V}_{\boldsymbol{X}}^- = \{i \in [n] \mid \mathcal{S}_i^{\text{norm}} < \tau_l\}.$$

In this case, the interval $[L_z^+, U_z^+]$ in (17) and $[L_z^-, U_z^-]$ in (18) are modified as follows:

$$[L_z^+, U_z^+] = \bigcap_{i \in [n]} [L_z^+(i), U_z^+(i)],$$

$$[L_z^+(i), U_z^+(i)] = \begin{cases} [\max(L'_z, f_i(\tau_u)), U'_z] & \text{if } \left(\beta_i > \beta^* \wedge i \in \mathcal{V}_{\boldsymbol{a}+\boldsymbol{b}z}^+\right) \vee \left(\beta_i < \beta^* \wedge i \notin \mathcal{V}_{\boldsymbol{a}+\boldsymbol{b}z}^+\right) \\ [L'_z, \min(U'_z, f_i(\tau_u))] & \text{if } \left(\beta_i < \beta^* \wedge i \in \mathcal{V}_{\boldsymbol{a}+\boldsymbol{b}z}^+\right) \vee \left(\beta_i > \beta^* \wedge i \notin \mathcal{V}_{\boldsymbol{a}+\boldsymbol{b}z}^+\right), \end{cases}$$

$$[L_z^-, U_z^-] = \bigcap_{i \in [n]} [L_z^-(i), U_z^-(i)],$$

$$[L_z^-(i), U_z^-(i)] = \begin{cases} [\max(L'_z, f_i(\tau_l)), U'_z] & \text{if } \left(\beta_i > \beta^* \wedge i \notin \mathcal{V}_{\boldsymbol{a}+\boldsymbol{b}z}^-\right) \vee \left(\beta_i < \beta^* \wedge i \in \mathcal{V}_{\boldsymbol{a}+\boldsymbol{b}z}^-\right) \\ [L'_z, \min(U'_z, f_i(\tau_l))] & \text{if } \left(\beta_i < \beta^* \wedge i \notin \mathcal{V}_{\boldsymbol{a}+\boldsymbol{b}z}^-\right) \vee \left(\beta_i > \beta^* \wedge i \in \mathcal{V}_{\boldsymbol{a}+\boldsymbol{b}z}^-\right), \end{cases}$$

where $\beta^* = \tau \beta_p + (1 - \tau)\beta_q$ for $p = \arg\min_i(\mathcal{S}_i)$ and $q = \arg\max_i(\mathcal{S}_i)$, and $f_i$ is defined as

$$f_i(\tau) = \frac{\tau c_p + (1 - \tau)c_q - c_i}{\beta_i - \beta^*}.$$

### D.3 Algorithm for Computing the Selective $p$-value

---

**Algorithm 1:** Computation of the selective $p$-value

---

**Input:** $G_{\boldsymbol{X}^{\mathrm{obs}}}$

**Output:** $p_{\mathrm{selective}}$

$\mathcal{Z} \leftarrow \emptyset$ ;

Obtain $\mathcal{V}_{\boldsymbol{X}^{\mathrm{obs}}}$ from $G_{\boldsymbol{X}^{\mathrm{obs}}}$ by (3) ;

Compute $\boldsymbol{a}$, $\boldsymbol{b}$ by Lemma 2 ;

Initialize $z$ to a sufficiently small value ;

**while** $z$ is not sufficiently large **do**

$\quad$ Compute $[L_z, U_z]$ and $\mathcal{V}_{\boldsymbol{X}(z)}$ by (16) for $z$ ;

$\quad$ **if** $\mathcal{V}_{\boldsymbol{X}(z)} = \mathcal{V}_{\boldsymbol{X}^{\mathrm{obs}}}$ **then**

$\quad\quad$ $\mathcal{Z} \leftarrow \mathcal{Z} \cup [L_z, U_z]$ ;

$\quad$ $z \leftarrow U_z + \gamma$, where $0 < \gamma \ll 1$ ;

Compute $p_{\mathrm{selective}}$ by (13) ;

**return** $p_{\mathrm{selective}}$ ;

---

# E   Evaluation of Robustness

In this experiment, we confirmed the robustness of the proposed method to two factors that may affect the Type I error rate control. First, we evaluate the robustness to non-Gaussian noise. Second, we evaluate the robustness to estimated variance.

## E.1   Robustness to Non-Gaussian Noise

In this experiment, we confirmed the proposed method can control the Type I error rate when the data is generated from a non-Gaussian distribution. As non-Gaussian noise, we considered the following five distribution families:

- **skewnorm**: Skew normal distribution family.

- **exponorm**: Exponentially modified normal distribution family.

- **gennormsteep**: Generalized normal distribution family (where the shape parameter $\beta$ is constrained to be steeper than that of the normal distribution, i.e., $\beta < 2$).

- **gennormflat**: Generalized normal distribution family (where the shape parameter $\beta$ is constrained to be flatter than that of the normal distribution, i.e., $\beta > 2$).

- **t**: Students t-distribution family.

Note that all of these distribution families include the Gaussian distribution and were standardized in the experiment.

To conduct the experiment, we first obtained a distribution such that the 1-Wasserstein distance from $\mathcal{N}(0, 1)$ was $\Delta$ for each distribution family, with $\Delta \in \{0.01, 0.05, 0.1, 0.15\}$. Other settings were the same as the default settings in the Type I error rate evaluation in Section 6.2. The results are shown in Figure 6 demonstrating that our proposed method can effectively control the Type I error rate for non-Gaussian noise.

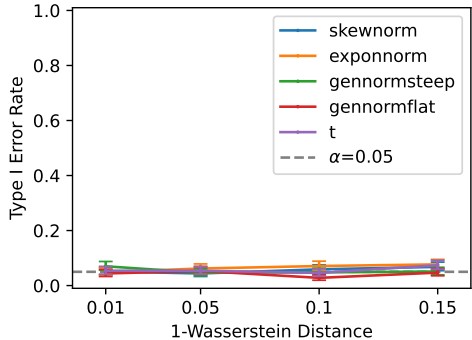

Figure 6: Type I error rate for non-Gaussian noise.

### E.2 Robustness to Estimated Variance

In this experiment, we confirmed that the proposed method can control the Type I error rate when the variance is estimated as sample variance from the same data. We varied the number of nodes $n \in \{32, 64, 128, 256\}$ and evaluated the Type I error rate at three significance levels: $\alpha = 0.01, 0.05, 0.10$. Other settings were the same as the Type I error rate evaluation in Section 6.2. The results are shown in Figure 7, demonstrating that our proposed method can effectively control the Type I error rate.

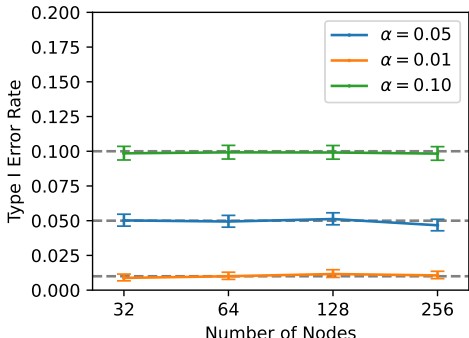

Figure 7: Type I error rate for estimated variance.

## F    Sensitivity of the Threshold Parameters

In this experiment, we confirmed that the proposed method can control the Type I error rate when the threshold $\tau_u$ in (1) and $\tau_l$ in (2) are varied. We varied the threshold pair $(\tau_l, \tau_u)$ over $\{0.1, 0.3, 0.5, 0.7\} \times \{0.2, 0.4, 0.6, 0.8\}$ such that $\tau_l < \tau_u$. Other settings were the same as the Type I error rate evaluation in Section 6.2. The results are shown in Figure 8, demonstrating that our proposed method can effectively control the Type I error rate at the $\alpha = 0.05$ level for all combinations of $\tau_l$ and $\tau_u$.

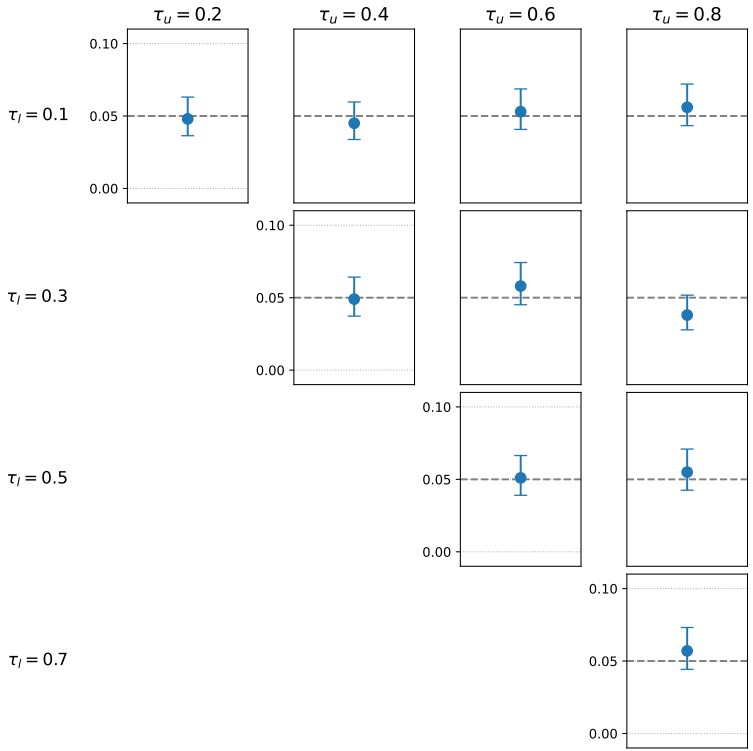

Figure 8: Type I error rate when varying the threshold parameters $\tau_l$ and $\tau_u$.

# G   Evaluation with Various GNN Architectures and Saliency Map Methods

In this experiment, we confirmed that the proposed method can control the Type I error rate across various combinations of GNN architectures and saliency map methods, which satisfy the assumptions of piecewise linearity. We compared the following four methods:

- **GCN+CAM**: This setting is the same as in Section 6, where the GNN architecture is GCN and the saliency map method is CAM.

- **GCN+GradCAM**: The GNN architecture is GCN and the saliency map method is Grad-CAM. We used the same 3-layer GCN as in Section 6 and applied Grad-CAM to the output of the second layer.

- **GIN+Grad**: The GNN architecture is GIN (Graph Isomorphism Network) and the saliency map method is Grad (Shrikumar et al., 2017). In this setting, we implemented GIN based on the standard formulation (Xu et al., 2019), where $v$-th node features $\boldsymbol{h}_v = \boldsymbol{X}_{v,:}$ at $l$-th layer are updated as follows in the GINLayer:

$$\boldsymbol{h}_v^{(l+1)} = \mathrm{MLP}\big((1+\epsilon)\boldsymbol{h}_v + \Sigma_{u \in \mathcal{N}(v)}\boldsymbol{h}_u\big),$$

  The MLP consisted of two layers with 64 hidden units and a 64-dimensional output, and used ReLU as the activation function. The parameter $\epsilon$ was learned. After three GINLayers, the node features were aggregated via global add pooling to obtain a graph-level representation, which was then passed through a fully connected layer to compute the logits for two classes.

- **GIN+GradInput**: The GNN architecture is GIN and the saliency map method is GradInput (Shrikumar et al., 2017). The GIN architecture is the same as in the `GIN+Grad` method.

All methods satisfied the assumptions of piecewise linearity. All saliency maps were normalized to the range $[0, 1]$ before applying thresholds to extract salient and non-salient subgraphs.

Other settings were the same as the Type I error rate evaluation in Section 6.2. The results are shown in Figure 9, demonstrating that our proposed method can effectively control the Type I error rate at the $\alpha = 0.05$ level for all settings.

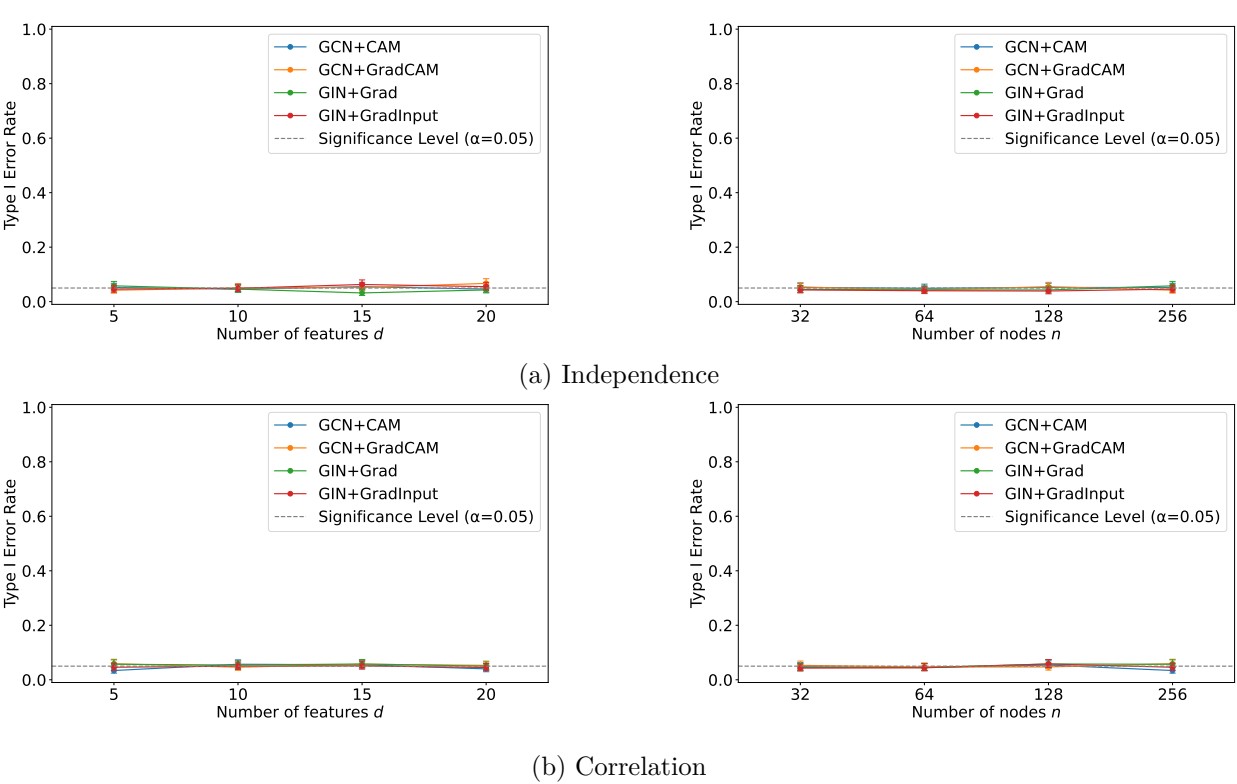

(a) Independence

(b) Correlation

Figure 9: Type I error rate for various GNN architectures and saliency map methods.

# H  Experiments on EEG Dataset

## H.1  Detail of Dataset

The EEG dataset provided by Won et al. (2022) consists of high-resolution brain activity recordings obtained from 55 participants performing the Rapid Serial Visual Presentation (RSVP) task. In this dataset, each trial comprises a sequence of visual stimuli presented at a rapid rate, with participants required to identify target stimuli among distractors. The EEG signals are labeled based on whether a stimulus was a target (positive category) or a non-target (negative category). The event-related potential (ERP) component of interest in this study is the P300 response, a well-established neural marker for target detection in RSVP paradigms.

The dataset includes signals collected from 32 electrodes positioned according to the international 10-20 system, sampled at 512 Hz. The original recordings spanned from -200 to 1000 ms relative to stimulus onset, capturing both pre-stimulus baseline and post-stimulus neural activity. Each participant's dataset consists of 40 positive and 560 negative samples.

The preprocessing and filtering steps adhered to standard EEG signal processing practices; for further details, refer to Won et al. (2022). However, specific modifications were made in this study to enhance data quality. To mitigate the influence of eye movement artifacts, four sensors near the eyes were excluded from analysis, leaving 28 sensors for further processing. Only the post-stimulus interval (0 to 1,000 ms) was analyzed, as the primary interest lies in stimulus-evoked activity rather than pre-stimulus baseline fluctuations. Each 1,000 ms segment was then downsampled from 512 Hz to 50 Hz, reducing the time series to 50 points while preserving the relevant frequency components associated with cognitive processing.

## H.2 Examples of Individual Results

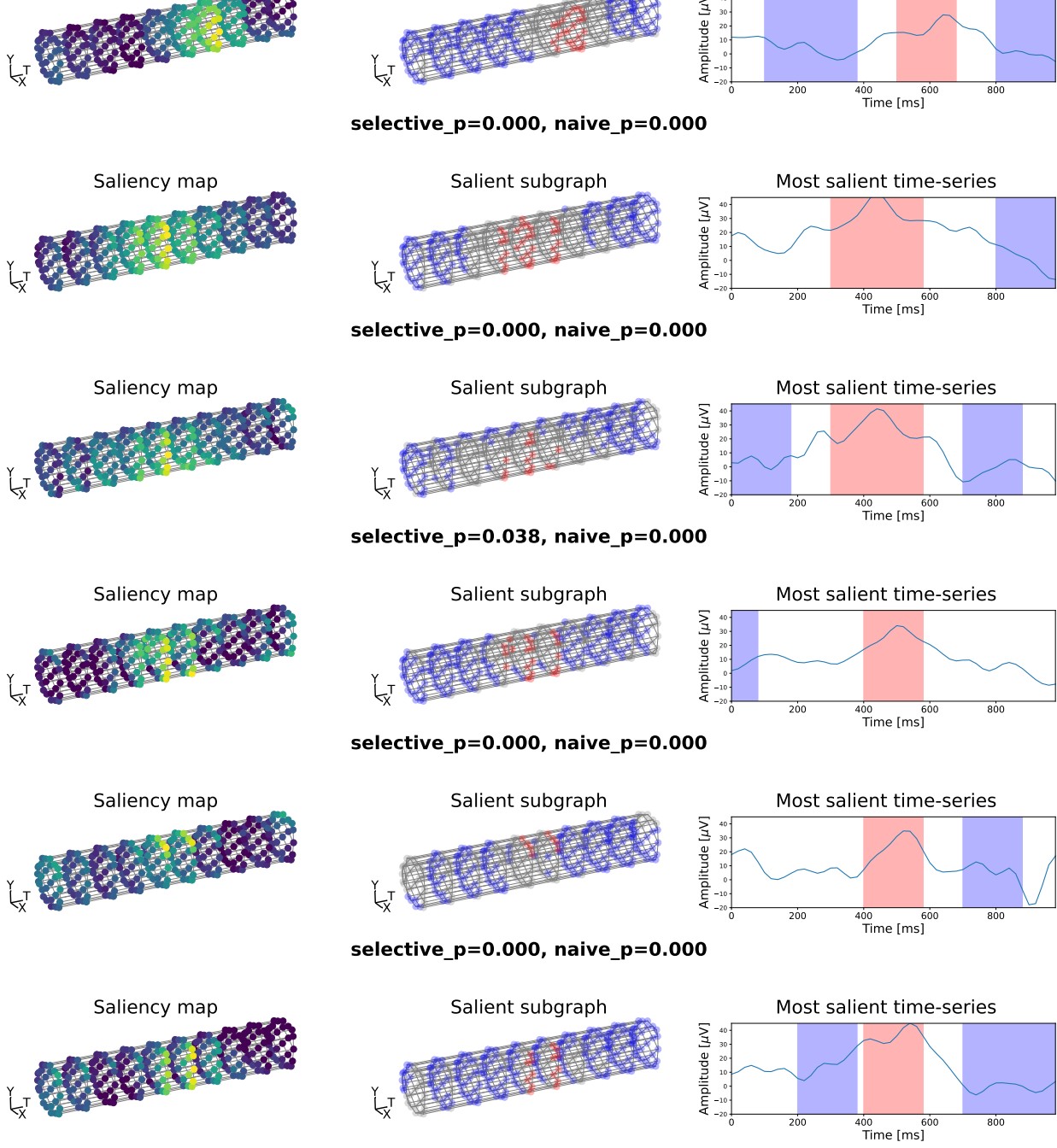

Figure 10: Positive examples. See Figure 4 for the interpretation of the visual elements. Below each example, we report the corresponding $p$-values, demonstrating that the proposed method correctly detects the positive samples.

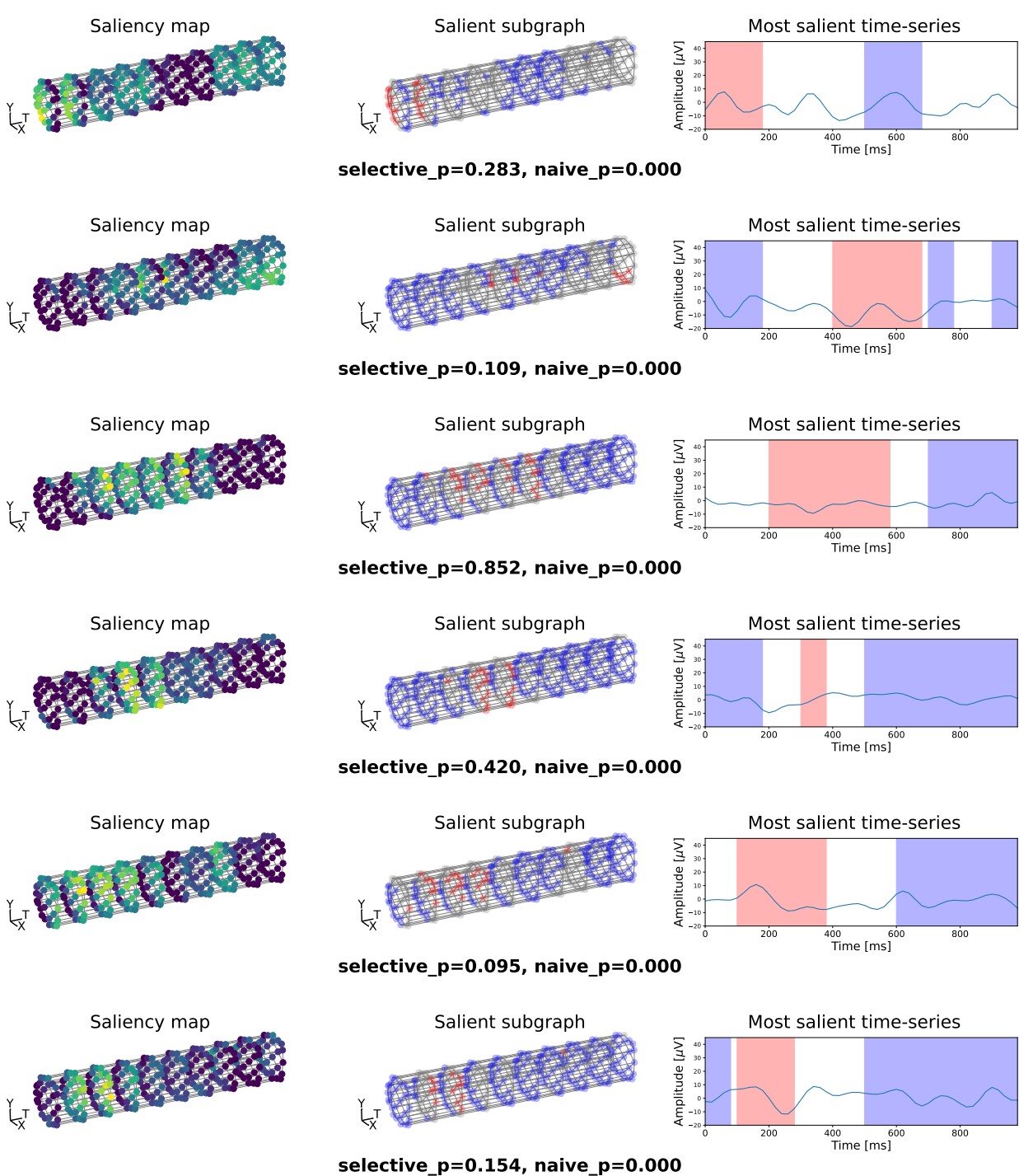

Figure 11: Negative examples. See Figure 4 for the interpretation of the visual elements. Below each example, we report the $p$-values for both the proposed and naive methods. The selective $p$-values are sufficiently large, indicating correct exclusion of spurious saliency, whereas the naive $p$-values are misleadingly small.

### H.3 Experiments on Modified Datasets

In this experiment, we evaluated the proposed method using modified real datasets.

The primary objective of this study is to develop a valid hypothesis testing framework under the assumption that the data follows the model presented in Section 3. However, real-world data do not always adhere to these assumptions, making direct evaluation challenging. In Section 6, we demonstrated several case studies to illustrate our approach.

In this section, we evaluate the Type I error rate by modifying the real datasets used in Section 6 to better conform to the assumed model. For the experiment, we estimated the mean vector $\mu^+$ from the positive class and the covariance matrix $\Sigma$ from the negative class in the real dataset. To investigate the effect of covariance structures, we considered two different settings for $\Sigma$: (i) the sample covariance matrix, normalized so that its largest eigenvalue is one and then scaled by a factor $\gamma \in \mathbb{R}$ (denoted as `full`), and (ii) a diagonal covariance matrix set to $\gamma I$ (denoted as `eye`). The scalar factor $\gamma$ was varied over $\{0.25, 0.5, 0.75, 1.0\}$. Using these estimates, we generated 1,000 test samples following $X \sim \mathcal{N}(\mathbf{0}, \Sigma)$ for Type I error rate evaluation and $X \sim \mathcal{N}(\mu^+, \Sigma)$ for power analysis.

The results are presented in Figure 12. The proposed method effectively controls the Type I error rate while achieving a detection rate above the significance level $\alpha$.

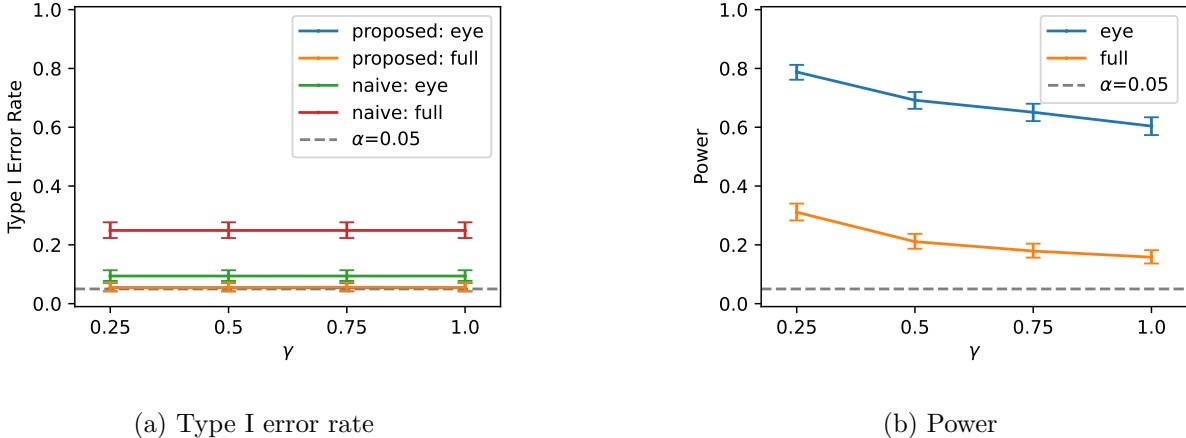

(a) Type I error rate                                          (b) Power

Figure 12: Results for modified real datasets

