# OpenReview forum: "Statistical Test for Saliency Maps of Graph Neural Networks via Selective Inference"
_TMLR — Accepted by TMLR_

### Review · Reviewer_DAiL · 2025-06-16

**Summary Of Contributions:**

The work presents a novel method to conduct statistical tests on saliency maps retrieved for a graph neural network. The goal of the test is to identify _salient_ sets of features compared _non-salient_ sets of features with statistical guarantees (limiting the Type-1 error rate). The statistical test proposed in this work is based on the framework of selective inference. The proposed test is compared and evaluated on synthetic experiments and applied on a real-world example.

**Audience:**

Yes

**Broader Impact Concerns:**

No concerns.

**Claims And Evidence:**

Yes

**Requested Changes:**

### Ablation Study with different GNN architectures
Conduct an ablation study as described in the weakness section (item 1.) similar to the experiment in Section 6.2 but with different GNN architectures and different saliency map methods. I am unsure about which of the mentioned methods (GAT, GIN, Grad-CAM, IG) would work for your test. If a method does not work for your test, it would be very important to describe why it does not. The results could be presented similarly as Figure 3.

### Better Plotting of Salient Time Series
I don't know how exactly this can be achieved, but it would be good, if the "evidence" in Figure 3 would be reworked. It is not clear to me if the salient sets provided in Section 6.3 are actually meaningful or not from an application perspective. Some further investigation and presentation in the form of a case study would be very appreciated. For a bit more context please see the weaknesses section item 2.


### Typos:
- Typo I noticed in your caption of Figure 3: I think it should be "The left and middle columns ..." instead of "The right and middle columns ...".
- Style: I do not like the "§" Symbol for cross-references inside the manuscript and would prefer writing "Section" or "Appendix".

**Strengths And Weaknesses:**

This is a good paper and I thank the authors for their work. I see a lot of positive aspects in this work and have only 2 concerns.

### Strengths:
- A very well written paper which is very easy to follow. Important aspects of the work are well presented in the main section and supplementary material is deferred to the Appendix.
- The contributions of the paper are very good. Statistical analysis of Explainable AI methods is well overdue since works like [1] emerged. Therein, the contributions of this work are very well positioned.
- Evidence for all claims made in the paper are provided  (with an exception of the ablation mentioned in the Weaknesses section).
- Strong related work section.
- Figures and Results are clearly stated.
- The theoretical results are clean and easy to follow (see my note).
- The Appendix is well structured and organized. It contains a lot of good additional information like the Computation of the p-values and the additional empirical evaluations like the robustness tests.

### Weaknesses:
1. While the paper shows that their proposed statistical test can be applied to any model / saliency combinations which adheres to the assumptions mentioned in Section 5. This could have easily been validated and demonstrated by an ablation study in the empirical evaluation. Currently, only one kind of architecture is used; a traditional GCN. It would be worth showcasing how other architectures like Graph Attention Networks (GATs) [2] or Graph Isomorphism Networks (GINs) [3] influence the statistical test. The same holds true for the choice of saliency method. Currently only CAM is applied, which is already quite dated. Better representatives in my opinion would be Grad-CAM or even integrated gradients (IG) [4]. I acknowledge that the work tightly limits its scope by specifically mentioning this as its main limitation, which seems to be in line with the guidelines provided by TMLR. However, the paper could be of much higher impact if these simple checks would be presented, since the title of the work currently reads as "Statistical Test for Saliency Maps of Graph Neural Networks via Selective Inference" **and not** "Statistical Test for _Class Activation Maps_ of Graph _Convolutional_ Neural Networks via Selective Inference".
2. The real-world experiment in section 6.3 falls a bit short without knowledge if the detected salient sets **actually are** salient sets, which are important. The plot with the most-salient time series might suggest that, but this is only a 1d view into the high-d time series. More evidence, that the test actually finds salient sets could be provided by maybe plotting not only the top-1 time series but maybe all dimensions in some meaningful manner or digging deeper into the application and assessing the results with domain experts or comparing the identified sets with other research from the medical domain.

### Note
I am not a statistician by trait and do not feel comfortable in judging the final correctness of all the theoretical results, since I might have overlooked an assumption which would be obvious to someone with more of a background in statistics.

### References
- [1] https://arxiv.org/abs/1810.03292
- [2] https://arxiv.org/abs/1710.10903
- [3] https://arxiv.org/abs/1810.00826
- [4] https://arxiv.org/abs/1703.01365

---

> ### Author Response · Authors · 2025-07-04
> **Our Replies to Reviewer DAiL (part 1)**
>
> We sincerely appreciate your time in reviewing our paper and providing constructive feedback. We especially thank you for careful reading and suggestions to improve the clarity of the writing.
>
> > Weaknesses 1. While the paper shows that their proposed statistical test can be applied to any model / saliency combinations which adheres to the assumptions mentioned in Section 5. This could have easily been validated and demonstrated by an ablation study in the empirical evaluation. Currently, only one kind of architecture is used; a traditional GCN. It would be worth showcasing how other architectures like Graph Attention Networks (GATs) [2] or Graph Isomorphism Networks (GINs) [3] influence the statistical test. The same holds true for the choice of saliency method. Currently only CAM is applied, which is already quite dated. Better representatives in my opinion would be Grad-CAM or even integrated gradients (IG) [4]. I acknowledge that the work tightly limits its scope by specifically mentioning this as its main limitation, which seems to be in line with the guidelines provided by TMLR. However, the paper could be of much higher impact if these simple checks would be presented, since the title of the work currently reads as "Statistical Test for Saliency Maps of Graph Neural Networks via Selective Inference" and not "Statistical Test for Class Activation Maps of Graph Convolutional Neural Networks via Selective Inference".
>
> The applicability of our method is limited to saliency methods in which the computation of saliency can be expressed as a piecewise linear function. This condition on piecewise linearity may seem restrictive, but we would like to emphasize that it is satisfied for most CNN-type networks with ReLU activation functions. In graph convolutional neural networks (GCNs), the fundamental building blocks—such as convolutional layers, ReLU activations, and max pooling—are all piecewise linear operations. As a result, the entire CNN can be considered a piecewise linear function, where the input space is partitioned into regions within which the network behaves linearly [1-3].
>
> [1] https://proceedings.neurips.cc/paper_files/paper/2014/file/fa6f2a469cc4d61a92d96e74617c3d2a-Paper.pdf
> [2] https://arxiv.org/abs/1606.05336
> [3] https://openreview.net/forum?id=B1J_rgWRb
>
> We have revised the manuscript to explicitly state this requirements of piecewise linearity in the abstract and in Section 1. We have also added a discussion in Section 5 (Limitations and Scope) about methods such as GATs and IG that do not satisfy the piecewise linearity.
>
> Furthermore, to clearly demonstrate that our proposed method can control the Type I error rate for a variety of configurations that satisfy piecewise linearity, we have added experimental results in Appendix G for configurations beyond GCN+CAM, specifically: GCN+GradCAM, GIN+Grad, and GIN+GradInput.
>
> > Weaknesses 2. The real-world experiment in section 6.3 falls a bit short without knowledge if the detected salient sets actually are salient sets, which are important. The plot with the most-salient time series might suggest that, but this is only a 1d view into the high-d time series. More evidence, that the test actually finds salient sets could be provided by maybe plotting not only the top-1 time series but maybe all dimensions in some meaningful manner or digging deeper into the application and assessing the results with domain experts or comparing the identified sets with other research from the medical domain.
>
> The EEG dataset used in Section 6.3 was chosen because it contains the well-known potential fluctuation P300, which is a highly recognized phenomenon in the relevant domain and thus suitable for validating our proposed method. The salient subgraph regions illustrated in Figure 4 (or Figure 8) align well with known P300 findings. The representative waveforms also exhibit the characteristic P300 shape. We have added explanations about this correspondence to Section 6.3.

---

> ### Author Response · Authors · 2025-07-04
> **Our Replies to Reviewer DAiL (part 2)**
>
> > Ablation Study with different GNN architectures: Conduct an ablation study as described in the weakness section (item 1.) similar to the experiment in Section 6.2 but with different GNN architectures and different saliency map methods. I am unsure about which of the mentioned methods (GAT, GIN, Grad-CAM, IG) would work for your test. If a method does not work for your test, it would be very important to describe why it does not. The results could be presented similarly as Figure 3.
>
> Please refer to our response to Weakness 1.
>
> > Better Plotting of Salient Time Series: I don't know how exactly this can be achieved, but it would be good, if the "evidence" in Figure 3 would be reworked. It is not clear to me if the salient sets provided in Section 6.3 are actually meaningful or not from an application perspective. Some further investigation and presentation in the form of a case study would be very appreciated. For a bit more context please see the weaknesses section item 2.
>
> Please refer to our response to Weakness 2.
>
> > Typo I noticed in your caption of Figure 3: I think it should be "The left and middle columns ..." instead of "The right and middle columns ...".
>
> We have corrected this in the revised manuscript.
>
> > Style: I do not like the "§" Symbol for cross-references inside the manuscript and would prefer writing "Section" or "Appendix".
>
> We have revised the manuscript to replace the "§" symbol with the word "Section" or "Appendix," as suggested.

---

> > ### Comment · Reviewer_DAiL · 2025-07-04
> > **Appreciate the Changes Made!**
> >
> > I thank the authors for addressing my concerns so clearly! The changes you made are very well executed. I particularly like your addition of the ablation in Appendix G and the clear description in the limitations section.
> >
> > **Minor Note:** The discussion of what architectures and saliency-method combinations are applicable is now only in the limitations section: Maybe one could also add a small _table_ somewhere in section 4 (or in the appendix) of well-established architectures and saliency methods that would be applicable and which are not applicable. This would help users as they can digest such a table quicker. Further, having this discussion mainly in the limitations section (like it is now) feels rather _defensive_, which I think is not needed: The framing could be "these kinds of things we can test with" in comparison to "these kinds of things we cannot really test with". Of course such a list/table could never be exhaustive and gets outdated rather quickly, but the ones discussed already would be a good start! If the authors feel like that this does not really improves the paper, I would also understand this, but I think the authors have a good understanding of the methods.

---

> > > ### Author Response · Authors · 2025-07-08
> > > **Response to Minor Note**
> > >
> > > Thank you for reviewing our updates and for your positive feedback, especially on the ablation study and the clarifications in the limitations section.
> > >
> > > We also appreciate your suggestion to include a table summarizing which architectures and saliency methods are applicable and which are not. We agree that such a table could help readers quickly grasp the scope, and we will consider adding it in the camera-ready version.
> > >
> > > That said, we are mindful that such a table might oversimplify some nuances. For example, the piecewise linearity depends on specific settings — it holds for GCNs with ReLU but not with sigmoid activations. If we can present it in a way that remains accurate while being clear, we would be happy to include it.
> > >
> > > Thank you again for the thoughtful suggestion.

---

### Review · Reviewer_uvL7 · 2025-06-17

**Summary Of Contributions:**

The paper proposes a statistical testing framework to evaluate the significance of saliency maps for explaining GNN decisions. The authors leverage the selective inference framework to overcome the issue of double dipping caused due to the use of the dataset for hypothesis generation and testing. The approach is limited to testing saliency maps from the CAM approach with a standard GNN architecture. Empirical evidence is provided to support the proposed approach.

**Audience:**

Yes

**Claims And Evidence:**

No

**Requested Changes:**

Please see the Weaknesses

**Strengths And Weaknesses:**

**Strengths:**
1. Overall the idea of statistical testing of GNN explanations is useful and has not been considered very rigorously in the literature. So in that sense the paper is a good addition.

**Weaknesses:**
1. The hypothesis are defined based on input feature values for the nodes identified in salient and non-salient partitions of the graph. What happens if the input nodes come without features and are just one-hot encoded? Even if the nodes come with input features why should saliency be defined only in terms of input node features and not also the structure? For example, two nodes having the same feature values might be in different substructures of the graph. One of the substructures might be more important than the other.

2. The connection to explainability/interpretability is missing. Once the statistical significance of salient subgraph is established how do we know that it is a reliable explanation in the sense that it is faithful to the model and the model actually used it to make the decision. In this regard I would suggest to the authors to relate their framework with existing explanation evaluation metrics.

3. To the best of my understanding, the framework  is in itself not new but its application to GNNs is. But currently the framework is simply applied to GNNs without accounting for the role of graph structure that is unique to the graph data and models. For instance Lemma 1 and Lemma 2 do not take into account of the effect of the graph structure at all.

4. It is also not clear how the threshold for obtaining the most salient subgraph are chosen. Imagine the case where all saliency values are very similar to each other. I think a very first test should be to check if the saliency values can be used to obtain a small salient subgraph for instance by checking the entropy of the the returned saliency distribution as proposed in https://ieeexplore.ieee.org/document/9866587 for evaluation sparsity of continuous explanations. A very high entropy would already mean that there is no real distinguishing information in the obtained salient masks . More details should be provided how the current thresholds are chosen.

5. The presentation of the paper can be improved with a bit more details about the statistical testing framework as the general audience from Graph learning might not have background in statistical testing. Specifically Lemma 2 which is the core of this framework needs to be elaborated a bit more of what the conditional data space would mean for the given graph data.

---

> ### Author Response · Authors · 2025-07-04
> **Our Replies to Reviewer uvL7 (part 1)**
>
> We sincerely appreciate your time in reviewing our paper and providing constructive feedback.
>
> > Weaknesses 1. The hypothesis are defined based on input feature values for the nodes identified in salient and non-salient partitions of the graph. What happens if the input nodes come without features and are just one-hot encoded? Even if the nodes come with input features why should saliency be defined only in terms of input node features and not also the structure? For example, two nodes having the same feature values might be in different substructures of the graph. One of the substructures might be more important than the other.
>
> We focused on attributed graphs with continuous features, considering use cases such as EEG analysis, as presented in our paper. We have revised the manuscript to explicitly state this requirement in Section 1.
>
> In addition, our method is applicable only to saliency methods that satisfy that can be represented as a piecewise linear function. We have revised the abstract and Section 1 to explicitly state this assumption. For example, GNNExplainer, which extracts salient structures themselves, does not satisfy piecewise linearity and thus our method cannot be applied to it. We have also added a discussion on this limitation regarding saliency methods in Section 5 (Limitations and Scope). This condition on piecewise linearity may seem restrictive, but we would like to emphasize that it is satisfied for most GCN-type networks with ReLU activation functions. In graph convolutional neural networks (GCNs), the fundamental building blocks—such as convolutional layers, ReLU activations, and max pooling—are all piecewise linear operations. As a result, the entire CNN can be considered a piecewise linear function, where the input space is partitioned into regions within which the network behaves linearly [1-3].
>
> [1] https://proceedings.neurips.cc/paper_files/paper/2014/file/fa6f2a469cc4d61a92d96e74617c3d2a-Paper.pdf
>
> [2] https://arxiv.org/abs/1606.05336
>
> [3] https://openreview.net/forum?id=B1J_rgWRb
>
> Furthermore, to clearly demonstrate that our proposed method can control the Type I error rate for a variety of configurations that satisfy piecewise linearity, we have added experimental results in Appendix G for configurations beyond GCN+CAM, specifically: GCN+GradCAM, GIN+Grad, and GIN+GradInput.
>
>
> > Weaknesses 2. The connection to explainability/interpretability is missing. Once the statistical significance of salient subgraph is established how do we know that it is a reliable explanation in the sense that it is faithful to the model and the model actually used it to make the decision. In this regard I would suggest to the authors to relate their framework with existing explanation evaluation metrics.
>
> The statistical significance discussed in this paper serves as a reliability measure from a different perspective than explainability or interpretability. As demonstrated in this paper, by using selective inference, it is possible to properly quantify the statistical significance of CAM, which is one of the methods for explainability. Since selective inference is not limited to CAM, we believe that selective inference can also be applied to quantify the statistical significance of other explanation methods for GNNs, as suggested by the reviewer, and we consider this an important direction for future work. For these discussions, we added the paragraph Interpretability and explainability in section 5 (scope and limitation).
>
> > Weaknesses 3. To the best of my understanding, the framework is in itself not new but its application to GNNs is. But currently the framework is simply applied to GNNs without accounting for the role of graph structure that is unique to the graph data and models. For instance Lemma 1 and Lemma 2 do not take into account of the effect of the graph structure at all.
>
> With all due respect, we are afraid that there may be a misunderstanding in this comment. Our method does take graph structure into account. Saliency maps are computed based on message passing that leverages graph structure, and our method controls the Type I error rate by conditioning the test statistic on the entire propagation process. Lemmas 1 and 2 provide the mathematical guarantees of this property.

---

> > ### Comment · Reviewer_uvL7 · 2025-07-07
> > **Clarification on conditional input data space**
> >
> > Thanks for your response. The conditional data space as in Lemma 2 is only defined for input node features and not for graph structure. Is that not correct?

---

> > > ### Author Response · Authors · 2025-07-08
> > > **Clarification on the Role of the Graph Structure in Lemma 2**
> > >
> > > Thank you for reviewing our response and for raising this follow-up question.
> > >
> > > It is correct that the adjacency matrix $A$ does not explicitly appear in the statement of Lemma 2. However, this does not imply that $A$ has no influence on the space $\mathcal{X}$. In fact, in Lemma 2, the space $\mathcal{X}$ is determined by the salient subgraph $V_{\mathbf{X}}$, and this subgraph is defined based on the saliency map $\mathcal{S}(G_{\mathbf{X}})$, which is itself a function of the graph data $G_{\mathbf{X}}=(A, \mathbf{X})$. Therefore,  $A$ plays an essential role in defining the structure of $\mathcal{X}$.

---

> ### Author Response · Authors · 2025-07-04
> **Our Replies to Reviewer uvL7 (part 2)**
>
> > Weaknesses 4-1. It is also not clear how the threshold for obtaining the most salient subgraph are chosen. Imagine the case where all saliency values are very similar to each other. I think a very first test should be to check if the saliency values can be used to obtain a small salient subgraph for instance by checking the entropy of the the returned saliency distribution as proposed in https://ieeexplore.ieee.org/document/9866587 for evaluation sparsity of continuous explanations. A very high entropy would already mean that there is no real distinguishing information in the obtained salient masks.
>
> Checking the validity of saliency based on whether it satisfies certain criteria is a very interesting direction. Our test correctly controls the TypeI error rate, even when a generated explanation lacks such criteria. Therefore, using it in parallel with these complementary metrics can lead to a more reliable assessment of interpretability. We have added a discussion of this point to the paragraph Interpretability and Explainability in Section5 (Scope and Limitations).
>
> > Weaknesses 4-2. More details should be provided how the current thresholds are chosen.
>
> We normalize saliency values to the $[0,1]$ range and then apply fixed thresholds ($\tau_l=0.3$ and $\tau_u=0.7$). We have added this explanation to Section 6. In our framework, selective inference is conducted while accounting for the thresholding process, ensuring control of the Type I error rate for any thresholds. To support this claim, we added a sensitivity analysis of Type I error rate with respect to thresholds in Appendix F. We also added a discussion in Section 5 about the need for selective inference to account for adaptive thresholding if thresholds are adjusted dynamically based on saliency distributions.
>
> > Weaknesses 5. The presentation of the paper can be improved with a bit more details about the statistical testing framework as the general audience from Graph learning might not have background in statistical testing. Specifically Lemma 2 which is the core of this framework needs to be elaborated a bit more of what the conditional data space would mean for the given graph data.
>
> In our setting, the conditional data space $\mathcal{X}$ is the subset of graph data that generates the same salient subgraphs as those of the observed data. We have added this clarification to the explanation of Lemma 2.

---

### Review · Reviewer_jNsC · 2025-06-20

**Summary Of Contributions:**

This paper addresses the issue of assessing the reliability of GNN saliency maps, which are used to model the interpretability of GNNs. Specifically, the authors propose a principled statistical testing based on Selective Inference, which quantifies the statistical significance of saliency maps by computing p-values. The proposed method leverages the piecewise linearity of GNN saliency methods (e.g., CAM) and reduces the high-dimensional conditional inference problem to a one-dimensional search problem. Extensive experiments on both synthetic and real-world EEG datasets demonstrate that the proposed method outperforms naive and Bonferroni-corrected baselines by achieving better Type I error control and higher statistical power.

**Audience:**

Yes

**Broader Impact Concerns:**

The authors could discuss how their proposed method affects the reliability of GNN in decision-making, e.g., recommender system and healthcare (e.g., the EEG example in the paper).

**Claims And Evidence:**

Yes

**Requested Changes:**

1.	How can the method be extended to handle soft attention weights or other non-linear saliency mechanisms?
2.	How sensitive are the selective p-values to the choice of thresholds? Would an adaptive thresholding strategy be possible?

**Strengths And Weaknesses:**

Strength
1.	The paper addresses an under-explored problem, i.e., the robustness in GNN interpretability by explicitly modeling and correcting for selection bias through Selective Inference.
2.	The paper is well-written, i.e., the motivation and method are clear and easy to understand.
3.	The paper provides extensive empirical evidence, including synthetic and real EEG datasets, demonstrating that the proposed method uniquely avoids false positives in negative-class saliency and maintains higher statistical power.

Weaknesses
1.	The method fundamentally relies on the piecewise linearity of CAM and cannot be directly applied to attention-based architectures or black-box saliency methods like GNNExplainer or GraphMask. Why the proposed method can assess the robustness of general GNN explanations methods needs to be clarified.
2.	The choice of saliency thresholds seems to significantly influence the subgraph definition and test outcome, yet the paper does not examine how sensitive the method is to these hyperparameters.

---

> ### Author Response · Authors · 2025-07-04
> **Our Replies to Reviewer jNsC**
>
> We sincerely appreciate your time in reviewing our paper and providing constructive feedback.
>
> > Weaknesses 1. The method fundamentally relies on the piecewise linearity of CAM and cannot be directly applied to attention-based architectures or black-box saliency methods like GNNExplainer or GraphMask. Why the proposed method can assess the robustness of general GNN explanations methods needs to be clarified.
>
> Our method is applicable only to saliency methods that whose computation process can be represented as a piecewise linear function. We have revised the abstract and Section 1 to explicitly state this assumption. In addition, we added a discussion in Section 5 (Limitations and Scope) about explanation methods such as GNNExplainer, which do not satisfy piecewise linearity and to which our method cannot be applied. This condition on piecewise linearity may seem restrictive, but we would like to emphasize that it is satisfied for most GCN-type networks with ReLU activation functions. In graph convolutional neural networks (GCNs), the fundamental building blocks—such as convolutional layers, ReLU activations, and max pooling—are all piecewise linear operations. As a result, the entire GCN can be considered a piecewise linear function, where the input space is partitioned into regions within which the network behaves linearly. [1-3]. To further clarify that our proposed method can control the Type I error rate across various methods that satisfy piecewise linearity, we have added experimental results in Appendix G for configurations beyond GCN+CAM, specifically: GCN+GradCAM, GIN+Grad, and GIN+GradInput.
>
> [1] https://proceedings.neurips.cc/paper_files/paper/2014/file/fa6f2a469cc4d61a92d96e74617c3d2a-Paper.pdf
>
> [2] https://arxiv.org/abs/1606.05336
>
> [3] https://openreview.net/forum?id=B1J_rgWRb
>
>
> > Weaknesses 2. The choice of saliency thresholds seems to significantly influence the subgraph definition and test outcome, yet the paper does not examine how sensitive the method is to these hyperparameters.
>
> Our framework performs selective inference while explicitly accounting for the thresholding process, thereby guaranteeing control of the Type I error rate regardless of the choice of threshold. We have clarified this point in Section 4. Additionally, to support this claim, we added sensitivity experiments on the Type I error rate with respect to threshold values in Appendix F.
>
> > Requested Changes 1. How can the method be extended to handle soft attention weights or other non-linear saliency mechanisms?
>
> Please refer to our response to Weakness 1.
>
> > Requested Changes 2-1. How sensitive are the selective p-values to the choice of thresholds?
>
> Please refer to our response to Weakness 2.
>
> > Requested Changes 2-2. Would an adaptive thresholding strategy be possible?
>
> In this paper, we proposed a framework that controls the Type I error rate for any fixed threshold. Our method is also applicable when the range of saliency values is normalized before applying the threshold, for cases where the range of the saliency distribution varies depending on the data.
>
> In contrast, when adaptively adjusting the threshold according to the characteristics of the saliency distribution, it is necessary to perform selective inference that accounts for the potential bias in threshold selection process.  An important future work is to introduce a new conditioning scheme, inspired by existing studies on selective inference for image segmentation, which discuss conditioning on thresholding processes that consider the distribution of pixel intensities in image data (see Figure 1 in [1]). Although we could not conduct experiments on this idea within the review period, we have added the above discussion to Section 5.
>
> [1] https://openaccess.thecvf.com/content_CVPR_2020/papers/Tanizaki_Computing_Valid_P-Values_for_Image_Segmentation_by_Selective_Inference_CVPR_2020_paper.pdf
>
> > Broader Impact Concerns: The authors could discuss how their proposed method affects the reliability of GNN in decision-making, e.g., recommender system and healthcare (e.g., the EEG example in the paper).
>
> Thank you for this suggestion. We have added a discussion of the broader impact of our method in the revised manuscript.

---

### Author Response · Authors · 2025-07-04
**Revised Manuscript**

We revised the paper based on the feedback from the three reviewers. The revised parts are marked in red. The main changes are as follows:

[1] To clarify the scope of applicability, we explicitly stated in the abstract and Section 1 that the proposed method works only with saliency-based techniques which can be represented as the piecewise linear function. We also added a discussion in Section 5 explaining why methods such as GNNExplainer and GATs fall outside this scope.

[2] To demonstrate that our test controls the Type I error rate for a wide range of other settings, we added Appendix G with new experiments on GCN+GradCAM, GIN+Grad, and GIN+GradInput, all of which satisfy the above piecewise linearity.

[3] To verify robustness with respect to the choice of threshold values, we conducted sensitivity experiments on the Type I error rate with varying thresholds and reported the results in Appendix F.

[4] We clarified in Section 4 that our selective inference framework accounts for the thresholding process and guarantees control of the Type I error rate for any fixed threshold.

[5] To explain the preprocessing pipeline, we noted in Section 6 that saliency values are normalized to the $[0,1]$ range before applying the fixed thresholds $\tau_l=0.3$ and $\tau_u=0.7$.

[6] To show how our test complements existing evaluation criteria, we discussed in Section 5 how combining it with metrics such as fidelity and consistency can yield more reliable explanations.

[7] To improve accessibility for readers unfamiliar with selective inference, we added a more detailed description of the conditional data space $\mathcal{X}$ in the explanation of Lemma 2.

[8] To provide stronger empirical evidence, we refined the real-world EEG experiment in Section 6.3 by showing that the detected salient regions align with known P300 characteristics and updating the corresponding figure captions.

[9] To address potential societal implications, we added a broader-impact discussion highlighting how the method can affect decision-making reliability in domains such as healthcare.

[10] To enhance readability, we corrected the Figure 3 caption (“right” → “left”) and replaced every occurrence of “§” with “Section” or “Appendix” throughout the manuscript.

---

### Decision · Action_Editor_8g3r · 2025-08-04

**Recommendation:** Accept with minor revision

**Additional Comments:**

After discussing with one of the reviewers, I expect the authors to put in some additional work in positioning their work in the interpretability literature more carefully, also beyond what was done in the revised version after the rebuttal. The reviewer argued that statistically significant saliency maps do not necessarily provide interpretability, which should be made more evident in the final version of the paper. The reviewer also asserts that the authors should put more work into explaining why their statistical framework is useful for the sake of interpretability, e.g., by positioning it as a "complementary evaluation metric."

**Audience:**

Yes

**Audience Explanation:**

Statistically testing the significance of saliency maps is interesting to sub-communities (e.g., XAI) within TMLR's audience.

**Claims And Evidence:**

Yes

**Claims Explanation:**

The contributions are not entirely novel; however, the proposed claims are sufficiently supported by experiments and theoretical results.